# A Simple yet Effective Model for Homology-Aware RNA Secondary Structure Prediction

## Abstract

Predicting RNA secondary structure is essential for understanding RNA function and developing RNA-based therapeutics. Despite recent advances in deep learning for structural biology, its application to RNA secondary structure prediction remains contentious. A primary concern is the control of homology between training and test data. Moreover, deep learning approaches often incorporate complex multi-model systems, ensemble strategies, or require external data. Here, we present the *RNAformer*, a scalable axial-attention-based deep learning model designed to predict secondary structure directly from a single RNA sequence without additional requirements. We demonstrate the benefits of this lean architecture by learning an accurate biophysical RNA folding model using synthetic data. Trained on experimental data, our model overcomes previously reported caveats in deep learning approaches with a novel homology-aware data pipeline. The RNAformer achieves state-of-the-art performance on RNA secondary structure prediction, outperforming both traditional non-learning-based methods and existing deep learning approaches, while carefully considering sequence and structure similarities.

## 1 Introduction

Ribonucleic acid (RNA) is a polymer of four nucleotides that plays a critical role in gene expression, protein synthesis, and epigenetic regulation (Morris & Mattick, 2014). The functionality of RNA molecules is intrinsically linked to their structure, which is determined by a hierarchical folding process, dictated by the formation of local geometries of the so-called secondary structure of RNA (Tinoco Jr & Bustamante, 1999). In addition to its impact on the final 3D shape, RNA secondary structures provide insights into RNA functions and can guide the design of RNA-based therapeutics and nanomachines (Kai et al., 2021).

While deep learning methods have achieved experimental accuracy in 3D protein structure prediction (Jumper et al., 2021; Lin et al., 2023; Abramson et al., 2024), RNA structure prediction remains challenging (Kretsch et al., 2023; Das et al., 2023). Specifically, capturing topologies and key secondary structure features appears difficult even for the current best model, *AlphaFold 3* (Bernard et al., 2024). The accurate prediction of an RNA's secondary structure directly from its sequence of nucleotides is still an important unsolved problem in computational biology (Bonnet et al., 2020).

Traditional computational methods for RNA secondary structure prediction minimize free energy using thermodynamic nearest-neighbor energy parameters, typically derived from wet-lab experiments (Delisi & Crothers, 1971; Tinoco et al., 1971). The currently most widely used algorithms, *mfold* (Zuker, 1989), *RNAfold* (Hofacker et al., 1994), and *RNAstructure* (Mathews et al., 1998) use dynamic programming to efficiently calculate these energy minimizations. More recently, learning-based approaches were developed to replace or improve estimates of the thermodynamic parameters (Do et al., 2006; Andronescu et al., 2007; 2010; Zakov et al., 2011).

Inspired by the success of deep neural networks in the field of protein contact map prediction (Hanson et al., 2018), *SPOT-RNA* was proposed, and deep learning entered the field of RNA secondary structure prediction (Singh et al., 2019). This class of approaches can represent an RNA structure as an adjacency matrix, surmounting the limitations of a restricted set of predictable base interactions of previous approaches and thus being able to predict non-canonical base pairs (Olson et al., 2019), pseudoknots (Staple & Butcher, 2005), and even base multiplets (Bhattacharya et al., 2019). The initial success led to additional deep learning-based models in the field (Chen et al., 2020; Sato

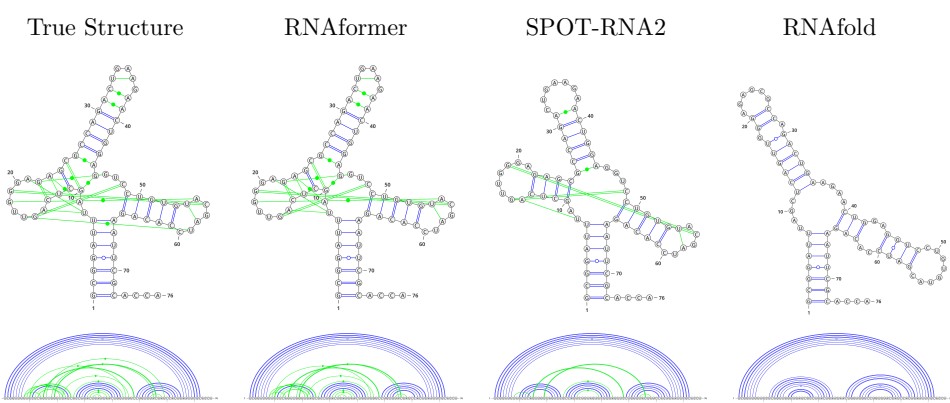

Figure 1: Example secondary structure predictions with different algorithms for a yeast tRNA[phe], the first RNA 3D structure determined by X-ray crystallography in the 1970s. (Deng et al., 2023)

et al., 2021; Franke et al., 2022; Fu et al., 2022; Chen et al., 2022; Chen & Chan, 2023), including the homology-based modeling approach, *SPOT-RNA2* (Singh et al., 2021b). This, however, has the disadvantage that it requires multiple sequence alignments (MSAs) of a sufficient size to achieve good predictions, which are not often available for RNAs (Singh et al., 2021b; Szikszai et al., 2022; Schneider et al., 2023; Bernard et al., 2024) resulting in poor generalization for orphan RNAs (Singh et al., 2021b; Bernard et al., 2024).

Despite the remarkable results reported by these approaches, deep learning methods are still not considered state-of-the-art for RNA structure prediction due to various problems (Flamm et al., 2021; Schneider et al., 2023; Das et al., 2023). One of these is their complexity; most methods require additional information such as MSAs (Singh et al., 2021b; Abramson et al., 2024) or thermodynamic parameters (Sato et al., 2021), build ensembles (Singh et al., 2019), depend on a fully trained foundation model (Chen et al., 2022), or use sophisticated pre- (Singh et al., 2021b) or post-processing (Fu et al., 2022) methods. Furthermore, many deep learning approaches are based on convolutional neural networks (CNN) Singh et al. (2019; 2021b); Sato et al. (2021); Fu et al. (2022), in which the receptive field of the input depends on the model depth Luo et al. (2016). This means a larger sequence requires a deeper network. While not a problem for data with a fixed-sized input such as pre-processed images, this may be problematic for RNA sequences which have varying lengths. Additionally, some models cannot be retrained due to undisclosed training pipelines (Singh et al., 2019; Flamm et al., 2021), making it difficult to reproduce reported results or build future models on top of these.

A major problem with previous deep learning-based approaches is that they did not sufficiently address homologies between the training and test data. This led to strong performance on homologous RNAs but poor generalization (Szikszai et al., 2022; Justyna et al., 2023; Qiu, 2023), which raised concerns in the community that the reported results were overly optimistic (Flamm et al., 2021). Typically, training and test data were split solely based on sequence similarity (Singh et al., 2019; Chen et al., 2020; Fu et al., 2022; Chen et al., 2022; Chen & Chan, 2023). However, it has been extensively reported that this method is not sufficient to remove all similarities, as functional RNAs are more conserved in structure than in sequence. For instance, it is well known that tRNAs fold into a conserved clover-leaf structure, while the sequence similarity is low (Szikszai et al., 2024).

In this work, we present the *RNAformer*, an axial-attention-based model for RNA secondary structure prediction from single sequence inputs. The attention mechanism itself is independent of the input size and the axial (row- and column-wise) attention architecture allows the RNAformer to process a 2D matrix in the latent space to directly model the adjacency matrix. Our model does not use additional features such as MSAs. While the single components of the architecture are not new, their usage for RNA secondary structure prediction is novel and outperforms existing approaches. In addition, an important contribution is our novel data curation pipeline. RNAs can be classified into so-called *RNA families*, based on their structure and sequence similarity. Thus, we use sequence- and structure-based alignments of RNAs to split our training and test data in a family-based manner; an essential step to reliably assess the performance of RNA secondary structure prediction methods

on unseen data. This allows for a clean split of training and test data, overcoming limitations in previous deep learning approaches for RNA secondary structure prediction.

We trained and evaluated the RNAformer in two settings. First, we study the scalability and learning capabilities of our model by learning a biophysical model of RNA folding. We address the problem of similarity and homology learning by training on a synthetic dataset and testing on unseen families. Our analysis reveals that the RNAformer can accurately capture the sequence and structure features while scaling linearly in the number of predicted base pairs, in contrast to the problematic quadratic scaling behavior in the number of predicted base pairs previously reported for deep learning approaches (Flamm et al., 2021). Second, we pre-train our model with a families-based train-test-split and use a fine-tuning strategy to evaluate on experimentally derived RNA secondary structures from the Protein Data Bank (PDB) (Berman et al., 2000). Here, the RNAformer shows superior performance on unseen PDB samples. In particular, our model seems to capture long-range tertiary interactions and base multiplets very well as exemplified in the prediction of PDB ID 1EHZ shown in Figure 1. These experiments demonstrate that the RNAformer outperforms other approaches, including those that use more sophisticated models, are trained on homologous data, or employ additional information such as MSAs. We make our non-homologous data splits, training pipeline, and pre-trained models publicly available[1].

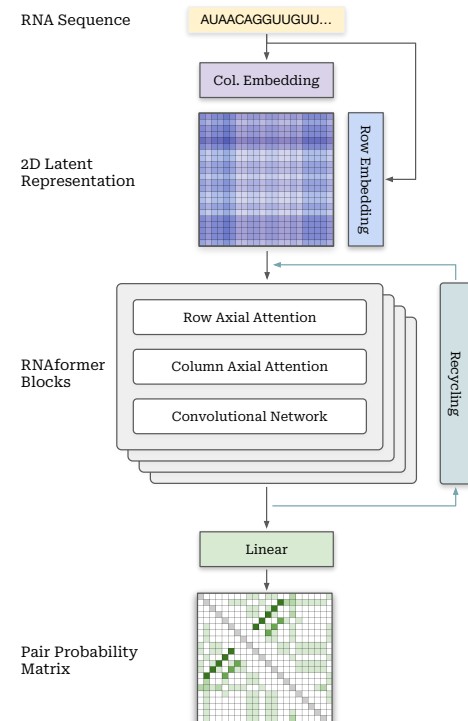

Figure 2: An overview of the neural architecture of the *RNAformer*. It has a lean design consisting of an embedding, axial-attention and convolutional network blocks, and a linear output layer.

In summary, our contributions are as follows:

- We present a lean, scalable, and interpretable deep learning architecture for RNA secondary structure prediction directly modeling the 2D pair matrix in latent space in Section 2.
- We introduce a novel data curation pipeline to overcome homology-related caveats raised by the RNA community in Section 3
- We provide an extensive experimental evaluation of both the model and data pipeline, and show state-of-the-art performance on RNA secondary structure prediction in Section 4

## 2 THE RNAFORMER

The architecture of the *RNAformer* is inspired by the protein folding algorithm *AlphaFold* (Jumper et al., 2021), which models a pair matrix in the latent space and processes it with the use of axial attention (Ho et al., 2019). In contrast to AlphaFold and similar to Lin et al. (2023), we dispense the use of a multi-sequence alignment due to its well-known limitations in RNA (Singh et al., 2021b; Schneider et al., 2023). In contrast to other deep learning based RNA secondary structure prediction models Singh et al. (2019; 2021a); Chen et al. (2022); Sato et al. (2021); Chen et al. (2020); Franke et al. (2022); Jung et al. (2022), we directly model the 2D pair matrix in latent space. This has the benefit, that the latent representation is already capable of representing pseudoknots and multiples and the final prediction of pairings between all nucleotides, the adjacency matrix, is just a linear operation. In contrast to purely CNN-based approaches Fu et al. (2022); Tan et al. (2024), our receptive field is not sequence length dependent.

---

[1]An anonymized repository is available at anonymous.4open.science/r/RNAformer_ICLR25.

## 2.1 Details of the Neural Architecture

The RNAformer (see Figure 2) inputs a nucleotide sequence $X \in \{A, C, G, U, N\}^l$ of length $l$ and embeds it twice, one row- and one column-wise embedding, to generate a 2D representation in the model's latent space. The embeddings can be represented as $E_{\text{row}} = \text{Embed}_{\text{row}}(X)$ and $E_{\text{col}} = \text{Embed}_{\text{col}}(X)$, where $E_{\text{row}} \in \mathbb{R}^{l \times d}$ and $E_{\text{col}} \in \mathbb{R}^{l \times d}$ are the row-wise and column-wise embeddings respectively, with $d$ being the latent dimension. The broadcasting and combination of these two matrices to form a 2D latent space can be represented as:

$$L^{(0)} = E_{\text{row}} \oplus E_{\text{col}}^T, \tag{1}$$

where $L^{(0)} \in \mathbb{R}^{l \times l \times d}$ is the 2D representation in the $d$-dimensional latent space, and $\oplus$ denotes the broadcasting and addition operation; i.e., $L^{(0)}[i,j] = E_{\text{row}}[i] + E_{\text{col}}^T[j]$. We use rotary position embedding instead of positional encoding Su et al. (2024). The resulting latent representation will be further processed by a stack of $M$ RNAformer blocks

$$L^{(i)} = \text{RNAformerBlock}(L^{(i-1)}), \quad \text{for } i = 1, 2, \ldots, M. \tag{2}$$

Each block consists of a row-wise and column-wise axial attention network *'AxialAttentionNet'*, followed by a transition network *'TransitionConvNet'*. We apply residual connections, pre-layer norm, and dropout to all three layers. A single RNAformer block can then be represented as:

$$\begin{aligned} L^{(i)'} &= L^{(i)} + \text{AxialAttentionNet}_{\text{row}}(L^{(i)}) \\ L^{(i)''} &= L^{(i)'} + \text{AxialAttentionNet}_{\text{col}}(L^{(i)'}) \\ L^{(i+1)} &= L^{(i)''} + \text{TransitionConvNet}(L^{(i)''}). \end{aligned} \tag{3}$$

An optimal attention for a 2D latent representation would create a 3D attention tensor, due to memory limitations, this is not feasible and we split the attention into two consecutive row- and column-wise *AxialAttentionNet*. Each *AxialAttentionNet* consists of a linear layer to create the query, key, and value and a linear layer to project its output. The axial attention mechanism (Ho et al., 2019) applies attention mechanisms over each axis independently, enabling memory-efficient multi-dimensional attention. More specifically, the axial attention mechanism can be mathematically represented with indices for rows $i$ and columns $j$ for each 2-dimensional input to the attention mechanism (Vaswani et al., 2017): query $Q \in \mathbb{R}^{l \times l \times d}$, key $K \in \mathbb{R}^{l \times l \times d}$, and value $V \in \mathbb{R}^{l \times l \times d}$ for a sequence length of $l$ and a latent dimension of $d$. We compute for each column $j = 1, \cdots, l$

$$\text{AxialAttention}_{\text{row}}(Q, K, V, j) = \text{softmax}\left(\frac{Q_{:,j,:}K_{:,j,:}^T}{\sqrt{d}}\right)V_{:,j,:}$$

and for each row $i = 1, \cdots, l$ the respective $\text{AxialAttention}_{\text{col}}(Q, K, V, i)$. Our model achieves a complete receptive field by applying attention consecutively along each axis, in contrast to purely convolutional networks (CNN) that expand this field over multiple layers. Therefore, in a CNN the number of layers required to achieve a full receptive field depends on the input length. This could be harmful for data with highly varied input lengths such as RNA sequences. Our approach may therefore be better suited for secondary structure prediction since each layer accesses the entire sequence and can iteratively refine the structure prediction.

The transition layer in the vanilla transformer is a point-wise feed-forward network. However, we found a convolutional network performs better in our architecture. The convolution helps to model local structures like stem-loops while the axial attention layers capture long-range information across the entire input structure. The *TransitionConvNet* consists of two convolutional layers with a SiLU activation function (Elfwing et al., 2018) in the middle.

To generate a prediction, we apply a single linear layer after the RNAformer stack and output the binary pairing probability matrix $P \in \mathbb{R}^{l \times l}$ of the secondary structure directly:

$$\text{P} = \text{sigmoid}(\text{Linear}(L^{(M)}))$$

This provides several advantages over the more commonly employed dot-bracket notation output (Hofacker et al., 1994; Franke et al., 2022), which in contrast, makes it impractical to predict multiplets, difficult to predict pseudoknots, and requires post-processing to create a pair matrix.

To artificially increase the model depth, we apply recycling in the latent space, similar to Jumper et al. (2021), allowing the model to reprocess and correct its own predictions internally: We apply the RNAformer blocks multiple times by normalizing and adding the block output back to the embedded input and then infer the RNAformer blocks again. During training, gradients are only computed for the last recycling iteration.

## 2.2 Sparse Adjacency Loss

The adjacency matrices representing RNA secondary structures tend to be heavily dominated by zero entries as an RNA sequence of length $l$ forms at most $l/2$ base pairs that result in $l^2 - l/2$ non-zero entries, without considering multiplets. We employ a masking technique during training to address the imbalance within such matrix representations. First, we mask everything except of all non-zero entries (all pairings) and a region around these non-zero entries. Next, we randomly select 40% of the remaining zero entries (80% during the fine-tuning stage) to exclude from the mask too, effectively utilizing approximately 40% (80%) of the adjacency matrix entries for training. We treat the prediction of each entry of the masked adjacency matrix as a separate classification problem, for which binary cross-entropy between prediction $P$ and the true value is calculated. The mean of the entry-wise losses is then minimized while the masked regions are ignored. Further details on masking are provided in Appendix A.

## 3 RNA Homology Aware Data Pipeline

A clean split between training and test data is crucial for the success of deep learning training and obtaining a reliable model. The responsible data pipeline consists of data collection, pre-processing, filtering, and splitting. For RNA, this data splitting has to account for both sequence and structure similarity to determine homology and evolutionary conservation (Rivas, 2021), as issues of homology contamination between training and test data are well-known (Rivas et al., 2012). In recent years, several deep learning models have been proposed for RNA secondary structure prediction, each providing state-of-the-art performance on various datasets (Singh et al., 2019; Chen et al., 2020; Fu et al., 2022; Chen et al., 2022; Chen & Chan, 2023; Jung et al., 2022; Tan et al., 2024). However, their results are often misleading due to flawed data processing pipelines that consider only sequence similarity, raising criticism and doubts within the community about the general capabilities of deep learning methods to learn the underlying RNA folding process (Flamm et al., 2021; Szikszai et al., 2022; Qiu, 2023). In this section, we outline our approach to preparing homology-aware RNA data splits. We first describe how we generate a synthetic dataset based on the notion of RNA families (Section 3.1), before we detail our strategy for splitting publicly available datasets into non-homologous subsets, considering both sequence and structure similarity (Section 3.2).

### 3.1 Family-Based Synthetic Data Generation

To test our architecture, we construct a independent, synthetic dataset based on selected RNA families from the Rfam database version 14.9. Specifically, we sample sequences from the covariance models of every RNA family and fold each of these sequences with RNAfold (Lorenz et al., 2011) to obtain the secondary structure. We assign the sequences of 30 families to the test set, 25 to the validation set, and the remaining 3796 families to the training set ensuring no overlap of families between the datasets. The resulting datasets contain 410408, 2727, and 3344 samples for training, validation, and testing, respectively. A more detailed description of the sampling process is provided in Appendix B.1.

### 3.2 Homology Aware RNA Data Splits

In the following, we explain our approach to curating an accurate, homology-aware RNA train/test split for experimental data that is independent of of the synthetic data used before.

**Data Collection** We collect a large training data pool from the following public sources: the bpRNA-1m meta-database (Danaee et al., 2018), the ArchiveII (Sloma & Mathews, 2016) and RNAS-trAlign (Tan et al., 2017) datasets provided by (Chen et al., 2020), all data from RNA-Strand (Andronescu et al., 2008), as well as all RNA-containing data from the Protein Data Bank (PDB) (wwp,

2019), downloaded on September 23, 2023. After removing redundant sequences, our initial data pool consists of $107,098$ samples. We use the commonly used test sets TS1, TS2, TS3, and TS-Hard, and the sets VL0, VL1, and another 50 randomly selected PDB samples for validation. All four test sets as well as VL0 and VL1 are originally provided by (Singh et al., 2019) and (Singh et al., 2021b). For our evaluations, we gather TS1, TS2, and TS3 into a single test set, *TS-PDB*, containing 125 samples, and keep the TS-Hard set separate for further analysis.

**Data Pre-processing** Secondary structures for PDB samples were derived from the 3D structure information using DSSR (Lu et al., 2015). For NMR-solved structures, such as those in TS2, model-1 structures were considered as the reference structure. For annotation of pseudoknots, we use bpRNA (Danaee et al., 2018), while ignoring base multiplets.

**Data Filtering** Our filtering pipeline consists of three steps: 1) We use CD-HIT-EST (Fu et al., 2012) to remove sequence similarity between training, validation, and test samples at the strictest threshold of $80\%$. 2) We perform a subsequent BLAST-N search (Altschul et al., 1997) to remove hits from the training and validation data for every test sequence at a very high e-value of 10. 3) We build covariance models, similar to those used for the RNA family database (Rfam) (Kalvari et al., 2020), for every test sample considering sequence and structure similarity using LocaRNA-P (Will et al., 2012) and Infernal (Nawrocki & Eddy, 2013), and remove every training and validation sample with a hit against any of the covariance models at an e-value of $0.1$. In line with the recent literature, we apply a general length cutoff at $500$ nucleotides to reduce computational costs during training (Singh et al., 2019; 2021b; Fu et al., 2022). For a more detailed description of this step, please see Appendix B.2.

**Data Splitting** For training the RNAformer, we curate three datasets: (1) Following the strictest data processing pipeline used in previous work (Singh et al., 2021b), we curate a pre-training dataset of $66,242$ samples that is non-homologous with respect to TS-Hard, while considering sequence similarity ($80\%$ similarity cutoff and BLAST-N search; see Appendix B.2) only for TS-PDB. This procedure ensures that the RNAformer is comparable to all other methods after pre-training. (2) To compare against AlphaFold 3, we create a non-filtered fine-tuning set, *FT-Homolog*, consisting of $4244$ samples drawn from all PDB entries in the initial pool, using the same cutoff date for data selection as AlphaFold 3 (September 30, 2021) without considering homologies but excluding sequences with an exact match in TS-PDB or TS-Hard. (3) Lastly, we create a fine-tuning set, *FT-Non-Homolog*, of $3432$ PDB samples without sequence and structure similarity between them and the test samples in TS-PDB and TS-Hard ensured by using our three-step-pipeline. We provide an overview of the datasets in Appendix Table B.2.

## 4 EXPERIMENTS

We perform two types of experiments: first, on synthetic data to test the architecture's capabilities (Section 4.1), and then on experimental data to show the impact of data homologies and performance on PDB samples ( Section 4.2).

All experiments are implemented in PyTorch (Paszke et al., 2019). We train the smallest RNAformer on a single NVIDIA A10 GPU and the others on 4-8 A100 GPUs. We use AdamW (Loshchilov & Hutter, 2019) with weight decay $0.1$ as the optimizer for pre-training and AdamCPR (Franke et al., 2023) with an $L_2$-norm constraint of $0.8$ of the initial parameter $L_2$ norm for fine-tuning. In both cases, we apply a cosine learning rate schedule and a learning rate warm-up. Due to the two-dimensional latent space, we have a higher memory footprint. We adress this by limiting the sequence length to $500$, using FlashAttention for a memory-efficient implementation (Dao et al., 2022), and using gradient accumulation. We provide additional hyperparameters in Appendix Table C. We treat RNA secondary structure prediction as a binary classification task over the base pairs and use the masking technique described in Section 2.2 to address the class imbalance.

### 4.1 LEARNING A BIOPHYSICAL MODEL

It was disputed in the RNA community whether deep learning approaches are generally capable of learning a biophysical model of RNA folding, or if the strong performance results from similarity between training and test samples only (Flamm et al., 2021; Szikszai et al., 2022; Qiu, 2023). We address this by learning a biophysical model implemented in the thermodynamic model of

Table 1: We train models in three different sizes, with (↻) and without recycling, on a dataset created with RNAfold predictions of the Rfam sequences. We evaluate on a family-based test set split and report the mean F1 score (see Appendix D) for the predicted base pairs across three runs with different random seeds. A structure is considered solved if the prediction matches the ground truth exactly. The scores increase with model size, demonstrating the scalability of the RNAformer architecture. High scores indicate that our model is capable of learning a biophysical model, similar to RNAfold.

| Model | Rfam TS | |
|---|---|---|
| | F1 score | Solved |
| RNAformer 32M ↻ | **0.967** | **83.5%** |
| RNAformer 32M | 0.948 | 68.1% |
| RNAformer 8M | 0.919 | 49.7% |
| RNAformer 2M | 0.846 | 22.9% |

RNAfold (Hofacker et al., 1994) as recently suggested (Flamm et al., 2021). A straightforward approach for our experiment is to use synthetic data, where we can generate large amounts of training samples while having full control over sequence length and family affinity as described in detail in Section 3.1. This simplified setup allows us to analyze different aspects of our model without restrictions on the number of available training samples and to assess the general ability of the RNAformer to learn reasonable features of RNA structures across families.

**RNAformer's Performance Scales with Model Size** An important property of modern deep learning algorithms is that their performance scales with data and model size (Kaplan et al., 2020). To test the scalability of the RNAformer, we train multiple RNAformer model sizes with $2M$, $8M$, and $32M$ parameters. We train each model three times with random seeds and report the mean results over the runs. As shown in Table 1, we increasingly approach RNAfold's "ground truth" results as we increase the model size. Our largest model achieves a high mean F1 score of $0.948$ ($\pm 0.026$; see Table 1 on the test set). This could be further improved by using latent space recycling (Jumper et al., 2021) to an F1 Score of $0.967 \pm 0.017$ and $84\%$ correct structure predictions.

**RNAformer Learns Sequence and Structure Features** To further analyze the structure predictions learned by the RNAformer, we compare its output to the ground truth data with respect to different structural features like stems or multi-loops and analyze the ratios of base pairs. The RNAformer captures all the structural features nearly perfectly with all base pair frequencies matched exactly, see Appendix Table E.1. In accordance with the underlying data generated with RNAfold, which only contains canonical base pairs, the RNAformer does not predict any non-canonical base pairs across three independent runs. Furthermore, our model strongly reduces false predictions of pseudoknots and multiplets. Previous work reported $48.8\%$ pseudoknot and $75.6\%$ multiplet predictions for the samples in a similar experiment without pseudoknots and multiplets in the datasets (Flamm et al., 2021). Here, only $1.9\%$ of the predictions of the RNAformer contained pseudoknots and $5.3\%$ included multiplet predictions.

It was recently shown that some deep learning methods scale quadratically in the number of predicted base pairs with increasing sequence length, a considerable flaw since a structure of length $l$ must form less than $l/2$ base pairs (ignoring multiplets) (Flamm et al., 2021). We, therefore, analyze the number of predicted base pairs of the RNAformer as a function of the sequence length when training on the synthetic data provided by RNAfold predictions. The results are summarized in Figure 3. Although the RNAformer slightly overestimates the number of base pairs, our analysis reveals robust support for linear scaling behavior and no heteroscedasticity of the residuals, we report details in Appendix F.

### 4.2 LEARNING HOMOLOGY AWARE RNA SECONDARY STRUCTURE PREDICTION

RNA secondary structures derived from 3D structures from PDB are considered the gold standard of secondary structure data. Most of the derived structures contain pseudoknots and base multiplets, which are typically excluded from most RNA secondary structure datasets and not considered by traditional methods. Since high-quality RNA data is scarce, recent deep learning methods typically

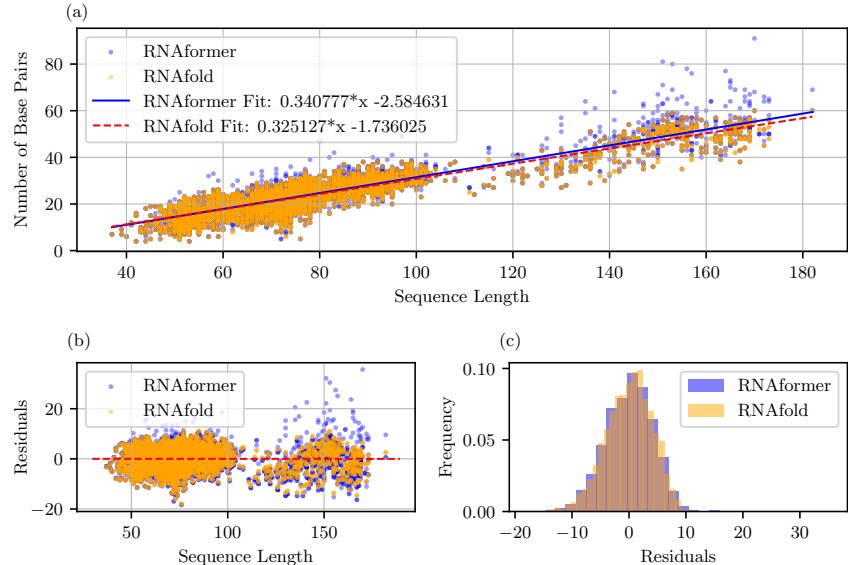

Figure 3: Analysis of the number of predicted base pairs of the RNAformer over different sequence lengths. a) Linear regression fit of the RNAformer predictions (blue) and the ground truth RNAfold (yellow). The number of predicted base pairs scales linearly with the sequence length for both the RNAformer predictions and RNAfold. b) Analysis of the residuals (errors). The residuals show no heteroscedasticity, albeit there are a few outliers for the RNAformer without statistical significance, indicating equal variance of the predictions over the sequence lengths. c) The distributions of the residuals. The error distributions of the linear regression models for RNAfold and RNAformer predictions appear visually similar and close to a normal distribution.

Table 2: The mean F1-score of the base pair predictions from three fine-tuning runs with different random seeds of the RNAformer compared to AlphaFold 3 on `TS-PDB` and `TS-Hard`.

| Model (trained without homology awareness) | TS-PDB | TS-Hard |
|---|---|---|
| RNAformer | **0.855** | **0.845** |
| Alphafold 3 (Abramson et al., 2024) | 0.817 | 0.688 |

use a fine-tuning strategy by first pre-training a model on large amounts of secondary structure data that was predominantly derived from comparative sequence analysis (Choudhary et al., 2017) and then fine-tuning the model on the high-quality experimental data (Singh et al., 2019; 2021b; Fu et al., 2022). We apply the fine-tuning strategy for the RNAformer by first pre-training a model on a large corpus of data collected from several publicly available databases (see Section 3) and then fine-tuning our model on high-quality experimental data collected from the PDB as described in Section 3. For evaluation, we use the two test sets `TS-PDB` and `TS-Hard`. Both test sets contain all types of base interactions, including non-canonical base pairs, pseudoknots, and base multiplets.

### 4.2.1 PREDICTION QUALITY WITHOUT HOMOLOGY AWARENESS

We first use the pre-trained model to finetune an RNAformer on the non-filtered dataset `FT-Homolog`. This experiment provides an upper bound for the performance by allowing homology between training and test data. We compare RNAformer with the current best RNA 3D structure prediction model, AlphaFold 3 (Abramson et al., 2024). This setup is a fair comparison because AlphaFold 3 did not consider data homologies either. Since Alphafold 3 predicts the entire 3D structure, we use DSSR (Lu et al., 2015) to extract the secondary structures from their tertiary predictions. The results comparing the RNAformer and AlphaFold 3 on RNA secondary structure prediction are shown in Table 2. Surprisingly, the RNAformer outperforms Alphafold 3 on both test sets, achieving high F1 scores of 0.855 and 0.845 on `TS-PDB` and `TS-Hard`, respectively, compared to 0.817 and 0.688 by AlphaFold 3. While the prediction of RNA 3D structures as done with

Table 3: The mean F1-score on the base pair prediction of three runs on `FT-Non-Homolog` with different randomly chosen seeds of the RNAformer in comparison to other methods on the `TS-PDB` and `TS-Hard` benchmarks. We use the following abbreviations for additional requirements: ES – Ensemble, TP – Thermodynamic Parameters, MSA – Multi-sequence Alignment, PRE – Pre-processing, PO – Postprocessing, EM – Embeddings. Only ss80 refers to methods with data pipelines that only consider sequence similarity at a cutoff of $80\%$ during data splitting. We observe that the RNAformer outperforms existing methods, despite having a stricter data pipeline and without additional data requirements such as MSAs. Please find additional metrics in Appendix G

| Model | Only ss80 | Additional Requirements | TS-PDB F1-Score | TS-Hard F1-Score |
|---|---|---|---|---|
| RNAformer finetuned | | – | **0.764** | **0.679** |
| RNAformer pretrain | | – | 0.723 | 0.601 |
| SPOT-RNA2 (Singh et al., 2021b) | | PRE, MSA | 0.754 | 0.666 |
| UFold (Fu et al., 2022) | ✗ | PO | 0.738 | 0.587 |
| SPOT-RNA (Singh et al., 2019) | ✗ | ES | 0.734 | 0.663 |
| RNA-FM (Chen et al., 2022) | ✗ | EM | 0.729 | 0.665 |
| MXFold2 (Sato et al., 2021) | | TP | 0.691 | 0.667 |
| ContraFold (Do et al., 2006) | | – | 0.669 | 0.625 |
| SPOT-RNA2 w/o MSA | | PRE | 0.668 | 0.637 |
| RNAFold (Lorenz et al., 2011) | n/a | TP | 0.659 | 0.636 |
| LinearFold-V (Huang et al., 2019) | n/a | – | 0.657 | 0.633 |
| IPknot (Sato et al., 2011) | n/a | PO | 0.652 | 0.611 |
| RNAstructure (Reuter & Mathews, 2010) | n/a | TP | 0.642 | 0.606 |
| LinearFold-C (Huang et al., 2019) | n/a | – | 0.632 | 0.610 |
| PKiss (Janssen & Giegerich, 2015) | n/a | TP | 0.615 | 0.613 |

AlphaFold 3 is arguably more challenging and typically would not allow a direct comparison, our results highlight that the RNAformer is capable of successfully learning the features of experimentally derived RNA secondary structures with high accuracy.

### 4.2.2 STATE-OF-THE-ART HOMOLOGY AWARE RNA SECONDARY STRUCTURE PREDICTION

We finetune the base model on the `FT-Non-Homolog` dataset to get a homology-aware RNA secondary structure prediction model. Table 3 provides an overview of results in comparison to other approaches. We report the mean performance across three training runs with different random seeds for the RNAformer. Despite other methods using less strict homology criteria in the data pipeline or additional requirements, they are outperformed by the RNAformer on the two test sets after fine-tuning with F1 Scores of 0.764 on `TS-PDB` and 0.679 on `TS-Hard`. We found the fine-tuning strategy crucial to achieving state-of-the-art results on experimental data. While solely pre-training achieved on-par performance with RNA-FM on the test set `TS-PDB`, fine-tuning further increased performance by roughly $6\%$. This improvement is even stronger for `TS-Hard`, where we observe an increase in the F1 Score of roughly $12\%$. We provide a visual comparison of the RNAformer to SPORT-RNA2 and RNAfold in Figure 1 and in Appendix Figure H.1. We find that the RNAformer captures the topology of the structure nearly exactly. We further analyze the predictions on `TS-PDB` as a function of the sequence length, we compare the RNAformer with the most commonly used methods. Appendix Figure I.1 shows the results of a linear regression fit for each of the models. The performance of the RNAformer only slightly decreases with the length of the input sequence, while we observe a stronger decrease RNAfold and RNAstructure.

Our evaluations on the test set `TS-Hard`, suggest that the performance of the RNAformer still correlates with the homology between training and test samples, since we apply the strictest data processing pipeline in the pre-training stage to `TS-Hard`. This is in line with the benefits from homologous data in the experiments in Section 4.2.1. However, our analysis of experimentally derived structures supports our finding that the RNAformer generalizes better to unseen RNA families. A key contributor to this achievement is our problem domain-specific architecture, which allows the RNAformer to successively refine its own prediction across layers, as indicated by the atten-

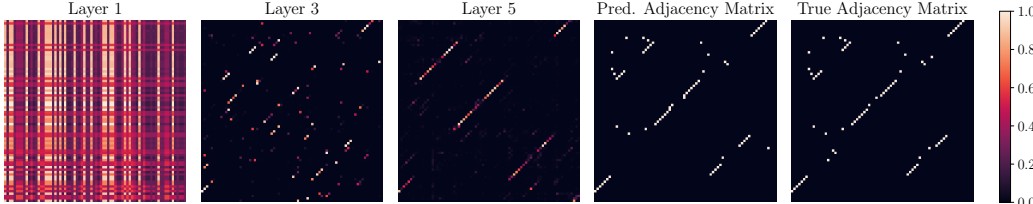

Figure 4: We analyze the attention matrices of different layers of the RNAformer, followed by its final predicted adjacency matrix, and the true structure for PDB ID 1EHZ (XRD, 1.93 Å). We observe a gradual refinement of the adjacency matrix in the latent space beginning with the row and column embedding in the first layer and gradually contributing to the final prediction. Each point in the final predicted matrix represents the probability that a base pair exists.

tion heatmaps shown in Figure 4. We provide additional plots of different samples in Appendix Figure J.1.

## 5 DISCUSSION & CONCLUSION

Our experiments show that the RNAformer architecture can learn a biophysical model as indicated by very high F1 Scores of $0.967$ when remodeling the predictions of RNAfold as well as nearly perfect reconstruction of the sequence and structure features (see Table 1 and Table E.1, respectively). Furthermore, our approach overcomes the quadratic scaling behavior in the number of predicted base pairs with increasing sequence length that was reported previously for deep learning-based approaches, scaling linearly in the number of predicted base pairs. Since we learn our model across families, we preclude that the strong performance results from learning data homologies, but rather, that it indicates RNAformer has learned the underlying biophysical model of the RNAfold modeling of the folding process.

Our study for the predictions on experimentally derived structures with all types of base interactions revealed that fine-tuning appears to be crucial to achieving strong performance. Compared to our pre-training model, the performance improved substantially after fine-tuning, resulting in state-of-the-art performance while using much stricter criteria to avoid homology between training and test data compared to previous work (Table 3). Notably, we observe that the RNAformer even outperforms the current state-of-the-art homology modeling method SPOT-RNA2 without the use of MSAs. The RNAformer is thus capable of predicting the structures of RNA for completely unseen families. The importance of the data processing is further supported by our strong performance on homologous data, outperforming AlphaFold 3.

On the downside, the additional performance of the RNAformer comes with a large memory requirement due to the two-dimensional representation in the latent space. However, an inference step is still very fast since it requires only one forward pass and advancements in tensor and graphic processing units increase the accessibility to larger deep learning models despite increasing computational demands. While the matrix representation of secondary structures enables deep learning methods to predict all types of base pairs, the sparsity of base pairs in the matrix bears the risk of incorrect classifications. The potential space of incorrect predictions scales quadratically with sequence length, while the number of base pairs only scales linearly. This achievement could be attributed to our loss masking technique which reduces the class imbalance. Nevertheless, in our experiments on synthetic data, the RNAformer strongly reduces the number of incorrectly predicted pseudoknots from roughly $50\%$ reported in previous work to roughly $2\%$ (Flamm et al., 2021). For multiplets, this reduction is even stronger where RNAformer predicts roughly $5\%$ of multiplets compared to $75\%$ reported previously (Flamm et al., 2021).

Overall, our comprehensive analysis clearly demonstrates that our lean deep neural network architecture achieves state-of-the-art performance on RNA secondary structure prediction without the need for additional features. The RNAformer overcomes the flaws reported for previous deep learning based approaches, making it a strong alternative to commonly used, non-deep learning based methods in the field.

## REPRODUCIBILITY STATEMENT

To ensure the reproducibility of our results, we have made our source code, model checkpoints, and datasets publicly available in the anonymous repository `https://anonymous.4open.science/r/RNAformer_ICLR25`. The repository contains detailed instructions for setting up the environment, including specific Python package versions (see `requirements.txt`). Model checkpoints for all experiments (base model, synthetic data trained model, and finetuned models) are provided in the `models` directory, with instructions for unpacking. Our datasets are stored in the `datasets` directory, also with decompression instructions. We provide scripts for evaluating trained models (`evaluate_RNAformer.py`) and for reproducing our training procedures (`pretrain_RNAformer.py` and `finetune_RNAformer.py`). The `README.md` includes specific command-line arguments for each experiment, ensuring exact replication of our results. Hardware requirements (GPU specifications) for both evaluation and training are clearly stated. By following the provided instructions, researchers should be able to reproduce our environment, evaluate our models, and replicate our experiments with minimal ambiguity.

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

APPENDIX

# A  MASKING

|        |        |        |
|:------:|:------:|:------:|
| (a)    | (b)    | (c)    |

Figure A.1: Mask construction process for a sequence of 23 nucleotides (nts) with padding of 2 due to a batch with a maximum length of 25 nts. (a) Original adjacency matrix with 4 base pairs (BPs). Black entries represent BPs, white-unpaired regions, and red entries indicate padding. (b) Mask after the first step (white area will be ignored during training): selecting regions with vicinity of 3 from the BPs. (c) Mask after second step: expanding mask to 60% of entries with random unpaired bases.

# B  DATA

This section details the process of dataset creation used for learning the RNA biophysical model as well as the homology-aware secondary structure prediction. In the latter case, we consider both sequence and structure similarity between the training and test sets to avoid data leakage due to structural homology, which is recognized as a significant factor in the model's performance. (Flamm et al., 2021; Szikszai et al., 2022; Qiu, 2023)

## B.1  SYNTHETIC DATASET

The synthetic dataset is created based on the Rfam database version 14.9 (Kalvari et al., 2020). First, we select Rfam families with covariance model (CM) characterized by the maximum number of matching position in the alignment (*CLEN*) of $\leq 500$ and generate a large set of sequences from CM of each selected family using Infernal. We then combine sampled sequences into the initial dataset in a way that guarantees two-thirds being generated from CMs with $CLEN \leq 200$ and one-third from CMs with $CLEN > 200$, with the latter used to increase family diversity. To obtain the corresponding secondary structures, the sequences are folded using *RNAfold*. Prior to that, we apply a length cutoff at 200 nucleotides because RNAfold's prediction accuracy drops on longer sequences and this reduces computational costs. The resulting dataset contains 410408, 2727, and 3344 samples from 3796, 25, and 30 families for the training, validation, and test sets, respectively. Further details can be found in Table B.1.

## B.2  EXPERIMENTAL DATASET

A common way of ensuring a fair comparison between RNAformer and its competitors would be to retrain and evaluate them on the same datasets. However, this would be computationally infeasible and in some cases impossible due to the undisclosed training pipelines (Singh et al., 2019; 2021b; Abramson et al., 2024). Hence, we propose an alternative strategy, in which we group methods based on the level of homology between their training and test sets defined by the sequence and structure similarity.

Recent publications report three different ways of assessing similarity between sets and following that, our data processing pipeline consists of three steps: 1) We use CD-HIT-EST (Fu et al., 2012) to remove sequence similarity between training, validation, and test samples, 2) a subsequent BLAST-N search (Altschul et al., 1997) removes hits from the training and validation sets for any sequence

in the test set, and 3) we implement a pipeline that considers sequence and structure similarity to ensure non-homologous data splits.

**Sequence Similarity**  We remove sequence similarity between the training, validation, and test sets by applying CD-HIT-EST with a similarity cutoff at $80\%$ between all sets. This pipeline is commonly used in previous works (Singh et al., 2019; Sato et al., 2021; Fu et al., 2022; Chen et al., 2022; Franke et al., 2022)

**BLAST-N Search**  In addition to removing similar sequences via CD-HIT-EST, we apply a BLAST-N-search (Altschul et al., 1997) at a high e-value of 10 to further remove training and validation samples that are recognized by BLAST-N as hits for any of the test samples. This pipeline was applied by SPOT-RNA and SPOT-RNA2 to further reduce sequence similarity (Singh et al., 2019; 2021b).

**Covariance Models**  We use BLAST-N (Altschul et al., 1997) to search for homologs for each sample of the test sets `TS-PDB` and `TS-Hard` using NCBI's nt database as a reference. We then create sequence- and structure-aware alignments using LocARNA-P (Will et al., 2012). For each of the resulting alignments, we build a covariance model using Infernal (Nawrocki & Eddy, 2013) and remove training and validation samples with a hit to the covariance model at an e-value of 0.1. A similar data pipeline was used for SPOT-RNA2 (Singh et al., 2021b), however, in that work the consensus structures for alignments were predicted using SPOT-RNA instead of an appropriate sequence- and structure-based alignment tool like LocARNA-P.

Table B.1: Overview of datasets used in the biophysical model experiment. This dataset is generated by inferring RNAfold.

| Dataset | # Samples | Length | | | | # Families |
|---|---|---|---|---|---|---|
| | | Minimum | Maximum | Mean | Median | |
| Train | 410408 | 22 | 200 | 95.2 | 85.0 | 3796 |
| Valid | 2727 | 34 | 160 | 80.2 | 78.0 | 25 |
| Test | 3344 | 37 | 182 | 79.4 | 74.0 | 30 |

Table B.2: Overview of datasets derived from experimental structures and comparative sequence analysis.

| | | Minimum | Maximum | Mean | Median |
|---|---|---|---|---|---|
| Pre-training | 66242 | 13 | 500 | 129.0 | 99.0 |
| FT-Homolog | 4244 | 4 | 200 | 57.9 | 47.0 |
| FT Non-Homolog | 3432 | 11 | 200 | 61.7 | 48.0 |
| Valid (Pre-Training) | 1302 | 33 | 497 | 131.0 | 105.0 |
| Valid (FT-Homolog) | 105 | 33 | 189 | 68.0 | 58.0 |
| Valid (FT Non-Homolog) | 35 | 33 | 159 | 76.4 | 64.0 |
| TS-PDB | 125 | 33 | 189 | 68.0 | 61.0 |
| TS-Hard | 28 | 34 | 189 | 65.6 | 50.5 |

## C  HYPERPAREMTERS

Table C.1: RNAformer Hyperparameters for Pretraining

| Hyperparameter | Homology-Aware Base Model | Biophysical Model |
|---|---|---|
| *Model Architecture* | | |
| Model Dimension | 256 | 256 |
| Number of RNAformer Blocks | 6 | 6 |
| Number of Attention Heads | 4 | 4 |
| ConvNet Dimension | 1024 | 1024 |
| ConvNet Kernel Size | 3 | 3 |
| Embedding Dropout | 0.4 | 0.1 |
| Residual Dropout | 0.4 | 0.1 |
| Layer Normalization Epsilon | 1.0e-05 | 1.0e-05 |
| Initializer Range | 0.02 | 0.02 |
| Maximum Sequence Length | 500 | 200 |
| Minimum Sequence Length | 10 | 10 |
| *Optimizer (AdamW)* | | |
| Learning Rate | 0.001 | 0.001 |
| Weight Decay | 0.1 | 0.1 |
| Beta 1 | 0.9 | 0.9 |
| Beta 2 | 0.98 | 0.98 |
| *Learning Rate Scheduler* | | |
| Schedule | Cosine | Cosine |
| Decay Factor | 0.01 | 0.01 |
| Warmup Steps | 1000 | 2000 |
| Total Training Steps | 20000 | 100000 |
| *Training Configuration* | | |
| Batch Token Size | 400 | 600 |
| Latent Recycling | 1 | 6 |
| Gradient Accumulation Steps | 8 | 0 |
| Number of Devices | 4 | 4 |
| Gradient Clipping Value | 1.0 | 1.0 |
| Number of Nodes | 2 | 2 |
| Precision | BF16 Mixed | BF16 Mixed |

Table C.2: RNAformer Hyperparameters for Finetuning

| Hyperparameter | Homology-Aware Finetuning | AF3-like Finetuning |
|---|---|---|
| Batch Size | 128 | 128 |
| Effective Batch Size | 4 | 4 |
| Maximum Sequence Length | 200 | 200 |
| Maximum Training Steps | 1200 | 4000 |
| Warmup Steps | 800 | 2000 |
| Learning Rate | 1.0e-06 | 1.0e-04 |
| Learning Rate Scheduler | Constant | Constant |
| Gradient Clipping Value | 0.1 | 0.1 |
| Number of Devices | 4 | 4 |
| Precision | BF16 Mixed | BF16 Mixed |
| Cycling | 8 | 8 |
| CPR Initialization | Dependent | Dependent |
| CPR Parameter | 0.8 | 0.8 |

## D    METRICS

We employ commonly used measures for RNA secondary structure prediction: F1 score, MCC, and F1-shift which are calculated based on a confusion matrix that describes the number of true positives (TP), true negatives (TN), false positives (FP), and false negatives (FN) of a given prediction.

**F1 score**    The F1 score describes the harmonic mean of precision ($PR = TP/(TP + FP)$) and recall ($RC = TP/(TP + FN)$ written as $F1 = 2 \cdot TP/(2 \cdot TP + FP + FN)$.

**Matthews Correlation Coefficient**    While the F1 score emphasizes positives, the MCC is a more balanced measure defined as:

$$MCC = \frac{(TP \cdot TN) - (FP \cdot FN)}{\sqrt{(TP + FP) \cdot (TP + FN) \cdot (TN + FP) \cdot (TN + FN)}} \tag{4}$$

**F1-shift**    The F1-shift accounts for structural dynamics in RNAs (Mathews, 2019) and is computed similarly to the F1 score with the difference that for a given pair $(i, j)$ all pairs $(i, j + 1)$, $(i + 1, j)$, $(i, j - 1)$, and $(i - 1, j)$ are also considered correct.

## E    QUALITATIVE ANALYSIS OF BIOPHYSICAL MODEL EXPERIMENT

Table E.1: Analysis of structural elements and base pair predictions of the RNAformer and the RNAfold algorithm. We use the following abbreviations: S – Stem, HL – Hairpin Loop, EL – External Loop, IL – Internal Loop, BL – Bulge Loop. The prediction frequencies of the elements are nearly identical.

| Relative frequency of bases in structural context | | | | | | |
|---|---|---|---|---|---|---|
| Model | S | HL | ML | EL | IL | BL |
| RNAformer | 0.602 | 0.132 | 0.016 | 0.089 | 0.109 | 0.030 |
| RNAfold Lorenz et al. (2011) | 0.607 | 0.131 | 0.015 | 0.090 | 0.105 | 0.031 |

| Relative frequency of base pair types | | | | | | |
|---|---|---|---|---|---|---|
| Model | AU | UA | GC | CG | GU | UG |
| RNAformer | 0.170 | 0.177 | 0.265 | 0.267 | 0.062 | 0.058 |
| RNAfold Lorenz et al. (2011) | 0.170 | 0.177 | 0.265 | 0.267 | 0.062 | 0.058 |

# F RNAFORMER'S NUMBER OF PREDICTED BASE PAIRS SCALES LINEARLY WITH SEQUENCE LENGTH

Using Ordinary Least Squares (OLS) regression, we observe that both the number of predicted base pairs of the RNAformer and RNAfold, scale linearly with sequence length (Figure 3 a)) with coefficients of 0.3251 and 0.3408 for RNAfold and the RNAformer, respectively. These results indicate a consistent increase in base pairs with longer sequences, albeit the RNAformer seems to slightly overestimate the number of base pairs as indicated by the slightly higher coefficient. The linear trend is further substantiated by high $\mathcal{R}^2$-values of 0.819 and 0.818 for the RNAformer and RNAfold, respectively, demonstrating a strong goodness of fit of the linear models, which explain a substantial portion of the variance in the data. Importantly, the residual analysis of the RNAformer predictions (Figure 3 b)) revealed no heteroscedasticity as indicated by a non-significant p-value of 0.152 using Levene's test.

Despite robust support for linearity, the residual distributions (Figure 3 c)) visually suggest normality; however, this hypothesis is statistically rejected as indicated by very low Shapiro-Wilk test p-values for the RNAformer as well as RNAfold (near zero). Typically, this deviation from normality could compromise the reliability of regression standard errors, influencing confidence intervals and hypothesis tests (Kutner et al., 2005). To address this, we employed a robust regression model using the Huber T norm with a MAD scale estimate, which reduces the influence of outliers and leverages points that could distort OLS regression estimations. This approach confirmed significant linear terms (coefficients of 0.3392 for RNAformer and 0.3284 for RNAfold), closely aligning with OLS regression results and bolstering our confidence in the linear model despite potential data anomalies.

Furthermore, we implemented a Generalized Linear Model (GLM) with a Poisson distribution, apt for modeling the count nature of base pairs which inherently accommodates their discrete and non-negative distribution. The Poisson GLM, with its log link function, facilitated a transformation aligning with the expected count data behavior, reinforcing a consistent linear relationship when back-transformed to the original scale (coefficients of 0.0109 for RNAformer and 0.0106 for RNAfold). The excellent model fit, indicated by pseudo $\mathcal{R}^2$ values exceeding 0.93, validates the linear trend across sequence lengths.

In conclusion, the comprehensive analysis employing OLS regression, robust regression, and Poisson GLM robustly confirms the linear scalability of the RNAformer's predicted number of base pairs with sequence length, providing a compelling alternative to previous models that were flawed by quadratic scaling.

# G  PERFORMANCE COMPARISON ON ADDITIONAL METRICS

To provide further insights into our evaluation, we list below the results of the different approaches in our comparison with more metrics from Section D.

Table G.1: The mean performance of three fine-tuning runs with different random seeds of the RNAformer in comparison to other methods on the `TS-PDB` benchmark.

| Model | TS-PDB | | | | |
|---|---|---|---|---|---|
| | F1-Score | F1-Shift | MCC | Precision | Recall |
| RNAformer finetuned | **0.764** | **0.793** | **0.767** | 0.834 | **0.720** |
| RNAformer pretrained | 0.723 | 0.750 | 0.733 | 0.846 | 0.655 |
| SPOT-RNA2 (Singh et al., 2021b) | 0.754 | 0.790 | 0.759 | 0.850 | 0.692 |
| UFold (Fu et al., 2022) | 0.738 | 0.770 | 0.741 | 0.816 | 0.686 |
| SPOT-RNA (Singh et al., 2019) | 0.734 | 0.758 | 0.742 | 0.851 | 0.665 |
| RNA-FM (Chen et al., 2022) | 0.729 | 0.752 | 0.741 | **0.878** | 0.643 |
| MXFold2 (Sato et al., 2021) | 0.691 | 0.718 | 0.704 | 0.856 | 0.593 |
| ContraFold (Do et al., 2006) | 0.669 | 0.697 | 0.678 | 0.803 | 0.588 |
| SPOT-RNA2 w/o MSA | 0.668 | 0.700 | 0.671 | 0.745 | 0.620 |
| RNAFold (Lorenz et al., 2011) | 0.659 | 0.682 | 0.667 | 0.792 | 0.576 |
| LinearFold-V (Huang et al., 2019) | 0.657 | 0.682 | 0.665 | 0.790 | 0.574 |
| IPknot (Sato et al., 2011) | 0.652 | 0.667 | 0.666 | 0.820 | 0.558 |
| RNAstructure (Reuter & Mathews, 2010) | 0.642 | 0.668 | 0.650 | 0.774 | 0.559 |
| LinearFold-C (Huang et al., 2019) | 0.632 | 0.650 | 0.648 | 0.809 | 0.537 |
| PKiss (Janssen & Giegerich, 2015) | 0.615 | 0.639 | 0.621 | 0.727 | 0.545 |

Table G.2: The mean performance of three fine-tuning runs with different random seeds of the RNAformer in comparison to other methods on the `TS-Hard` benchmark.

| Model | TS-Hard | | | | |
|---|---|---|---|---|---|
| | F1-Score | F1-Shift | MCC | Precision | Recall |
| RNAformer finetuned | **0.679** | **0.703** | **0.684** | 0.762 | **0.631** |
| RNAformer pretrain | 0.601 | 0.629 | 0.616 | 0.747 | 0.537 |
| MXFold2 (Sato et al., 2021) | 0.667 | 0.695 | 0.676 | 0.800 | 0.588 |
| SPOT-RNA2 (Singh et al., 2021b) | 0.666 | 0.700 | 0.673 | 0.757 | 0.620 |
| RNA-FM (Chen et al., 2022) | 0.665 | 0.691 | 0.683 | **0.845** | 0.575 |
| SPOT-RNA (Singh et al., 2019) | 0.663 | 0.686 | 0.674 | 0.796 | 0.593 |
| SPOT-RNA2 w/o MSA | 0.637 | 0.667 | 0.639 | 0.692 | 0.613 |
| RNAFold (Lorenz et al., 2011) | 0.636 | 0.659 | 0.641 | 0.752 | 0.561 |
| LinearFold-V (Huang et al., 2019) | 0.633 | 0.659 | 0.638 | 0.749 | 0.558 |
| ContraFold (Do et al., 2006) | 0.625 | 0.659 | 0.636 | 0.756 | 0.557 |
| PKiss (Janssen & Giegerich, 2015) | 0.613 | 0.640 | 0.620 | 0.715 | 0.558 |
| IPknot (Sato et al., 2011) | 0.611 | 0.617 | 0.624 | 0.767 | 0.528 |
| LinearFold-C (Huang et al., 2019) | 0.610 | 0.630 | 0.628 | 0.796 | 0.514 |
| RNAstructure (Reuter & Mathews, 2010) | 0.606 | 0.633 | 0.611 | 0.722 | 0.533 |
| UFold (Fu et al., 2022) | 0.587 | 0.623 | 0.591 | 0.657 | 0.553 |

# H ADDITIONAL SAMPLES OF RNA SECONDARY STRUCTURE PREDICTIONS

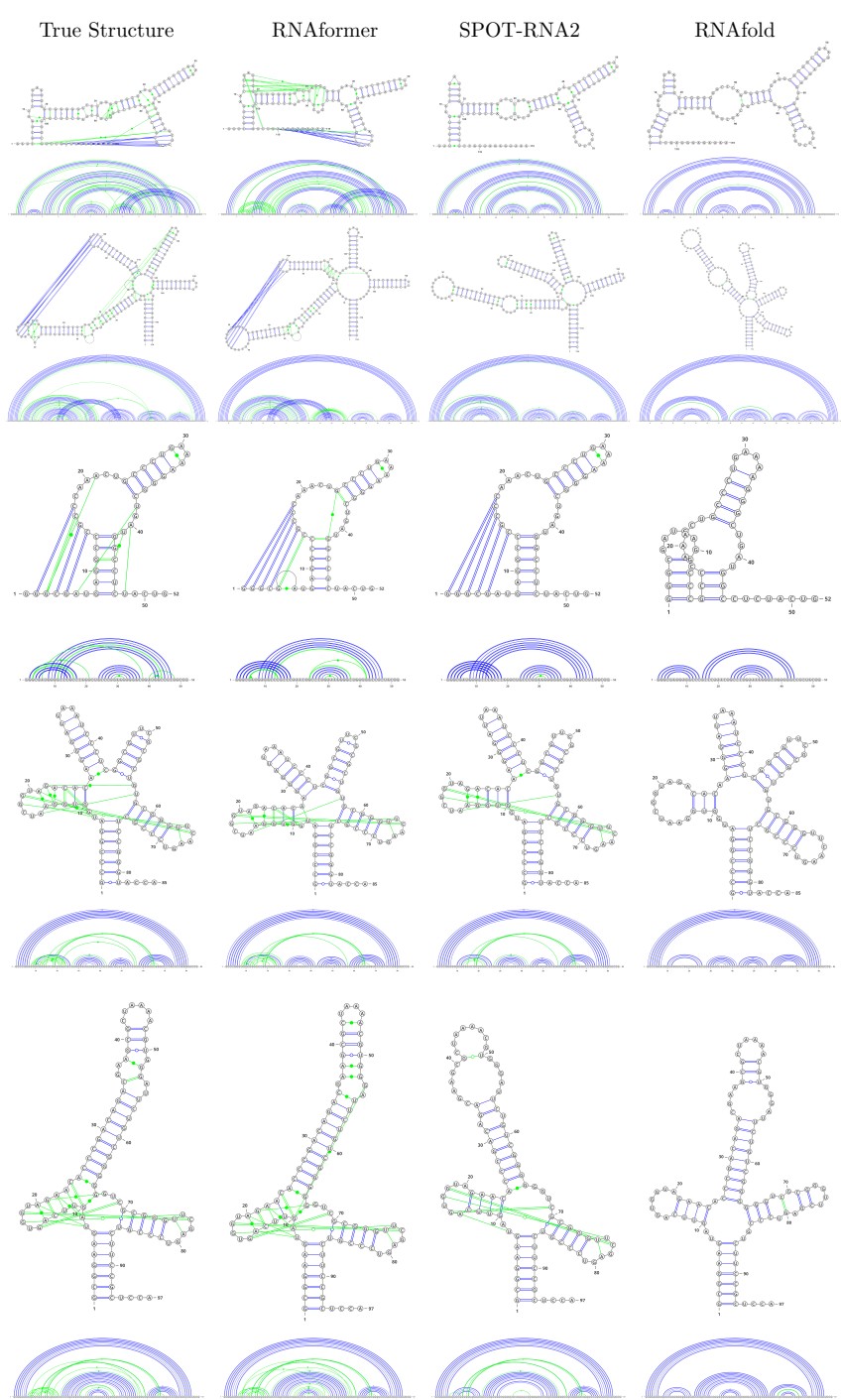

Figure H.1: Example secondary structure predictions for RNA samples with PDB IDs (top to bottom): 2LHP (NMR, 100%, E-Value: 2.264e-15), 2N6S (NMR, 100%, E-Value: 8.362e-15), 6PMO (XRD, 2.657 Å), 2NQP (XRD, 3.5 Å), and 4WJ3 (XRD, 3.705 Å). We find that the RNAformer captures the topology of the structure nearly exactly.

# I PERFORMANCE ANALYSIS OVER SEQUENCE LENGTH

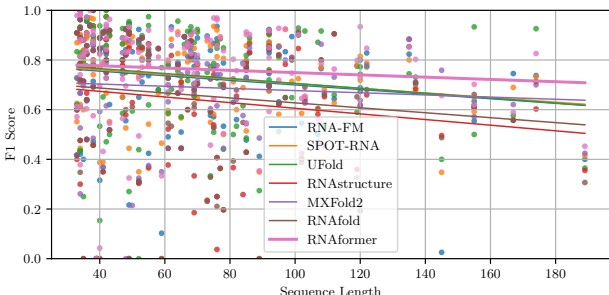

Figure I.1: Analysis of the base pair predictions of the RNAformer fine-tuned on PDB data (mean over three random seeds) in comparison to those of the competitors. The plot shows the F1 Scores of the single sequences in the TS-PDB test set over the sequence lengths. The line shows a linear regression fit and indicates that the performance of the RNAformer scales well with the sequence length.

## J    ADDITIONAL ATTENTION MATRIX FIGURES

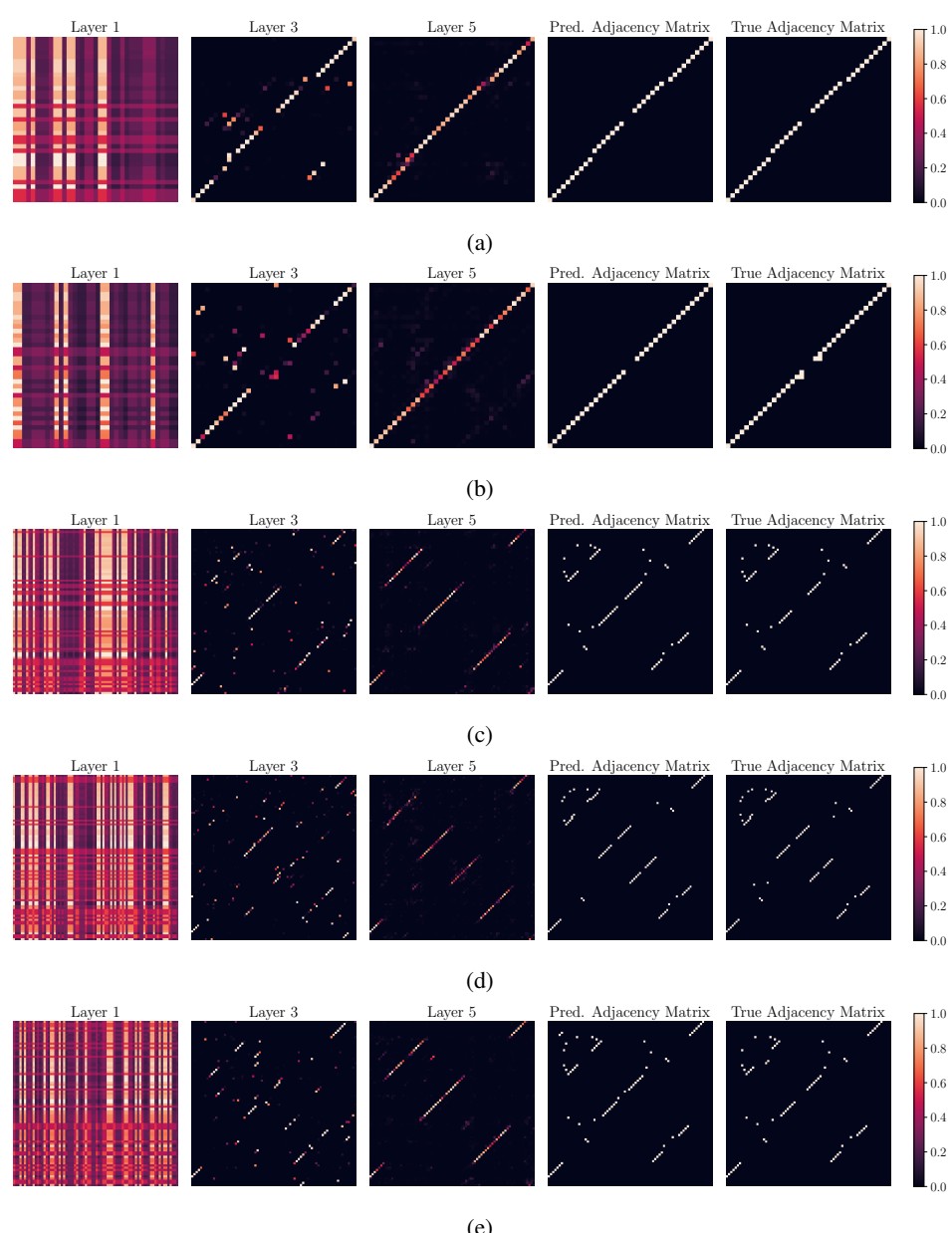

Figure J.1: Attention matrices of RNAformer predictions for RNA samples with PDB IDs: (a) 2LHP (NMR, 100%, E-Value: 2.264e-15), (b) 2N6S (NMR, 100%, E-Value: 8.362e-15), (c) 6PMO (XRD, 2.657 Å), (d) 2NQP (XRD, 3.5 Å), and (e) 4WJ3 (XRD, 3.705 Å). The first 3 matrices show the attention matrix of an attention head through RNAformer layers 1, 3, and 5. The final predicted adjacency matrix is then shown, followed by the adjacency matrix of the true structure.

