# OpenReview forum: "RNAformer: Axial-Attention For Homology-Aware RNA Secondary Structure Prediction"
_ICLR.cc/2025/Conference — Submitted to ICLR 2025_

### Official Review · Reviewer_UEd4 · 2024-11-02

**Soundness:** 3
**Presentation:** 3
**Contribution:** 2
**Rating:** 5
**Confidence:** 3

**Summary:**

The paper introduces RNAformer, a deep learning model that predicts RNA secondary structures from sequences using axial-attention mechanisms. It achieves state-of-the-art performance and overcomes previous limitations by employing a homology-aware data pipeline, making it superior to traditional and existing deep learning methods.

**Strengths:**

RNAformer stands out for its scalability, efficiency with varying sequence lengths, and a robust homology-aware approach to data curation. It simplifies the prediction process without additional data requirements and offers linear scaling capabilities, outperforming other models in accuracy and generalization.

**Weaknesses:**

1. In the introduction, why is it stated that AlphaFold 3 struggles to capture RNA topologies and secondary structure features? The article should provide a more detailed explanation. Since structural information is more conserved, why can't we solely use structure information for training and test data splitting?

2. What is meant by "row- and column-wise embedding" on page 4? According to the mathematical expression, $L^{(0)}$ is a 3D matrix; why is it described as being in a 2D latent space? The mathematical representation and textual explanation in this section are quite confusing, making it difficult for readers to understand.

3. From lines 156, 174, 179, and line 188, it appears that the strategies used in this article are derived from existing literature. Therefore, this work seems to be incremental in the context of RNA structure prediction. Where does the unique innovation of this article lie?

4. From a methodological standpoint, how does the author emphasize the issue of homology in data splitting?

5. The code repository provided in the article is inaccessible. Could the authors please provide an updated link for the code and data, so that readers can review and reproduce the results?

6. The deduplication tools should be CD-HIT-EST and BLAST-N.

7. Add the evaluation results for the INF (Interaction Network Fidelity) metric. Reference for INF: Parisien et al. RNA (2009), 15:1875–1885.

8. Why was the RNAFold-generated structure chosen as the true label for the synthetic dataset? According to later test results, there are other energy minimization-based prediction methods (e.g., MXfold2) that perform better.

**Questions:**

See "Weaknesses".

---

> ### Author Response · Authors · 2024-11-15
> **Response to Reviewer UEd4**
>
> Dear Reviewer UEd4,
>
> Thanks for the valuable feedback. We will address all your concerns and provide answers to questions in the following:
>
> >**Weaknesses:**
>
> >1. In the introduction, why is it stated that AlphaFold 3 struggles to capture RNA topologies and secondary structure features? The article should provide a more detailed explanation. Since structural information is more conserved, why can't we solely use structure information for training and test data splitting?
>
> Q1: “In the introduction, why is it stated that AlphaFold 3 struggles to capture RNA topologies and secondary structure features? The article should provide a more detailed explanation.”
>
> A: This part of the manuscript refers to the result in the cited publication (see [1]). C. Bernard et al. (2024) [1] recently analyzed the behavior of AF3 for RNA predictions, finding that specific predictions for orphan RNAs, as well as correct predictions for stacking and non-canonical base interactions, appear challenging for AF3.
>
> Q2: “Since structural information is more conserved, why can't we solely use structure information for training and test data splitting?”
>
> While RNA is typically more conserved in structure than in sequence, sequence similarity is still a valuable measure of similarity between two RNA sequences. Even if sequence similarity alone is not enough to ensure non-homologous data, it is thus still required for data processing pipelines.
>
> >2. What is meant by "row- and column-wise embedding" on page 4? According to the mathematical expression, L^0 is a 3D matrix; why is it described as being in a 2D latent space? The mathematical representation and textual explanation in this section are quite confusing, making it difficult for readers to understand.
>
> A vanilla transformer model works on a 1D input which is often a sequence of tokens and the latent space is a 2D-tensor (LxD with L sequence length and D latent dimension). Since we model a 2D matrix, the adjacency matrix of the RNA secondary structure, we also name the latent space 2D even though it's actually a 3D tensor (LxLxD with L sequence length and D latent dimension). So the 2D refers to the input-depended dimensions.
>
> We mean with "row- and column-wise embedding” an outer product of two independent embeddings of the AGCU-sequence into the latent dimension D. So $E_{row} = W_{row} \times onehot(inputseq)$, $E_{col} = W_{col}  \times onehot(inputseq)$ and $embed_{latent} = E_{row} \otimes E_{col} $. We can clarify this in the paper.

---

> ### Author Response · Authors · 2024-11-15
>
> >3. From lines 156, 174, 179, and line 188, it appears that the strategies used in this article are derived from existing literature. Therefore, this work seems to be incremental in the context of RNA structure prediction. Where does the unique innovation of this article lie?
>
> We see the innovation of our work as twofold. First, we prove that a simple axial-attention-based model is capable of achieving SOTA performance on RNA secondary structure prediction. No other work has used axial-attention for RNA secondary structure prediction. We show that there is no need for MSA, ensembles of architectures, additional features or constraints, or sophisticated pre- and post-processing.
>
> To achieve this, we develop an architecture that can deal with various lengths and is independent of the input length since we consider RNA sequence lengths from 33 to 200nt. The common usage of CNN-based architectures  (see e.g. [4]) are suboptimal since the receptive field depends on the input size, which is also true for U-Net architectures (see e.g. [5,11]). Only self-attention can deal with arbitrary input lengths while maintaining an input-independent full receptive field.
>
> Furthermore, we focus on the prediction of the adjacency matrix of base pairs because only the adjacency matrix representation can capture both, pseudoknots and multiplets (which appeared in real-world data in over 50% of the RNAs). To do so, we decided to have an architecture with a 2D latent representation (3D tensor with LxLxD shape with Length, and Dimension) that subsequently refines the adjacency matrix in the latent space. This also increases the interpretability of our model since we already model the final output format in the latent space, see Figure 4 and Appendix J.
>
> Since we can’t use fully CNN-based architectures as mentioned before, we have to use attention. A full attention on a 2D latent space would require a 3D attention matrix which is infeasible in terms of memory requirements. Therefore we disentangle the attention in a row- and column-wise attention similar to the pair matrix processing in the Evoformer (AlphFold2).  However, convolutions are well suited to capture local geometries. Therefore, one novel aspect of the RNAformer architecture is that it uses a single 3x3 convolutional layer instead of a feed-forward MLP (e.g. in AlphaFold2) to capture local structures (line 179). Another problem-specific novelty is the Sparse Adjacency Loss which we describe in Section 2.2 and is crucial for the SOTA performance.
>
> As our second major contribution, we present a novel data processing pipeline that ensures no data homologies between training and test sets. Recently multiple groups with biological backgrounds (see [2,8,9]) named caveats about the usage of deep learning in RNA structure biology. The main concern is insufficiently curated test data. In particular, the usage of test data sets that are not based on experimentally validated RNA structures and homology unaware splits between train and test data cause skepticism and harm the reliability and practical relevance of the developed models.
>
> For instance, training and evaluating on the RNAStralign and ArchiveII datasets as provided by Chen et al. 2020 [12] was shown to lead to overfitting and bad generalization capabilities (see [13]), and the bpRNA based TR0 and TS0 split as provided by Singh et al., 2019 [10], are known for homologies (see [2]). Nevertheless, we recently observed many approaches published at the latest AI conferences and top-tier journals that do not consider a rigorous data split, such as:
> - RFold (ICML 2024) [11]
> - E2Efold (ICLR 2020) [12]
> - RTfold (WCB@ICML2022) [6]
> - Probabilistic Transformer (NeurIPS 2022) [7]
> - SPOT-RNA (Nature Communications) [10]
> - UFold (NAR) [5]
> - RNA-FM (arXiv) [14]
> - (AlphaFold3 (Nature 2024); 3D prediction, yes, but still)
>
> We provide the first evaluations of a deep learning method for RNA secondary structure prediction on a non-homologous data split for experimentally validated structures.
>
> This sets a new standard for RNA data processing which allows deep learning practitioners to thoroughly evaluate their methods. We consider this a major contribution to the community and think that this is exactly the next step needed to bring deep-learning-based RNA structure prediction into production. Does this clarify your concerns regarding the innovation?

---

> ### Author Response · Authors · 2024-11-15
>
> >4. From a methodological standpoint, how does the author emphasize the issue of homology in data splitting?
>
> The issue of homology in RNA data splitting can be explained on a simple hypothetical example:
>
> A: GGAAAACC ((....))
>
> B: CCAAAAGG ((....))
>
> While the sequences A and B share the same secondary structure (in dot bracket notation here), only 50% of the nucleotides are the same. From an evolutionary point of view, one could argue that the evolutionary pressure is on the structurally relevant positions 1,2 and 7,8 since a change from G to C at position 1 seems to have triggered a change from C to G at position 8 during evolution to maintain the structurally relevant base pair between positions 1 and 8 (or similarly between 2 and 7). This would be easily observable in an MSA and such patterns could also be easily leveraged by a strong deep learning method in the case where we do not consider homology between the two sequences but only rely on sequence similarity measures. However, we would like to develop models that learn the folding of RNA and generalize to previously unseen samples instead of just recalling recurring patterns that have been learned during training. It is, therefore, important to consider both sequence and structure similarity to avoid homologous training and test data.
>
> To account for this problem, we split our data using a three-step approach. First, we remove similar sequences between training, validation and test data using CD-HIT-EST with a cutoff at 80% sequence similarity. This is a commonly (often solely) used approach in recent publications. In the second step, we apply BLAST-N at a high e-value of 10 to remove any sample from the training and validation sets that hit any of the test sequences during Blast search. Finally, we build covariance models for all test samples using Infernal based on sequence and structure-aware alignments obtained from LocaRNA-P. We then remove training and validation samples that have a hit with any covariance models at an e-value of 0.1. This stage is similar to the process of family assignments used for preparing the Rfam database.
>
> We describe the data pipeline in Section 3.2 and provide further details in Appendix B.2. Would this answer your question?
>
> >5. The code repository provided in the article is inaccessible. Could the authors please provide an updated link for the code and data, so that readers can review and reproduce the results?
>
> For us, the link in the paper works fine. Please find the code and datasets under: https://anonymous.4open.science/r/RNAformer_ICLR25/README.md
>
> >6. The deduplication tools should be CD-HIT-EST and BLAST-N.
>
> We thank the reviewer for this comment and we will correct this in the manuscript accordingly.
>
> >7. Add the evaluation results for the INF (Interaction Network Fidelity) metric. Reference for INF: Parisien et al. RNA (2009), 15:1875–1885.
>
> Please find tables with additional metrics in Appendix Table G.1. where we report the MCC score. According to Parisien et al. RNA (2009) the interaction network fidelity (INF) between structures A and B” is defined “as the MCC, INF(A,B) = MCC(A,B)”. For our evaluations, we mainly follow a recent publication [3] and further report the shifted F1 score.
> >8. Why was the RNAFold-generated structure chosen as the true label for the synthetic dataset? According to later test results, there are other energy minimization-based prediction methods (e.g., MXfold2) that perform better.
>
> We agree with the reviewer that we could have used other folding algorithms as well. This experiment, however, was performed in response to [2] where it is explicitly suggested to use RNAfold as a baseline. Furthermore, a different baseline does not add new insights since we only want to assess the general capabilities of the RNAformer architecture to learn a biophysical model of RNA folding.
>
> We hope we have clarified your questions and would be happy to answer any further questions. If we have addressed your concerns, we would be very thankful if you considered raising your score.

---

> ### Author Response · Authors · 2024-11-15
>
> ___
> [1] Bernard, C., Postic, G., Ghannay, S., & Tahi, F. (2024). Has AlphaFold 3 reached its success for RNAs?. bioRxiv, 2024-06.
>
> [2] Flamm, Christoph, et al. "Caveats to deep learning approaches to RNA secondary structure prediction." Frontiers in Bioinformatics 2 (2022): 835422.
>
> [3] Mathews, D. H. (2019). How to benchmark RNA secondary structure prediction accuracy. Methods, 162, 60-67.
>
> [4] Saman Booy, M., Ilin, A., & Orponen, P. (2022). RNA secondary structure prediction with convolutional neural networks. BMC bioinformatics, 23(1), 58.
>
> [5] Fu, Laiyi, et al. "UFold: fast and accurate RNA secondary structure prediction with deep learning." Nucleic acids research 50.3 (2022): e14-e14.
>
> [6] Jung, Andrew J., et al. "RTfold: RNA secondary structure prediction using deep learning with domain inductive bias." The 2022 ICML Workshop on Computational Biology. Baltimore, Maryland, USA. 2022.
>
> [7] Franke, Jörg, Frederic Runge, and Frank Hutter. "Probabilistic transformer: Modelling ambiguities and distributions for RNA folding and molecule design." Advances in Neural Information Processing Systems 35 (2022): 26856-26873.
>
> [8] Szikszai, Marcell, et al. "Deep learning models for RNA secondary structure prediction (probably) do not generalize across families." Bioinformatics 38.16 (2022): 3892-3899.
>
> [9] Qiu, Xiangyun. "Sequence similarity governs generalizability of de novo deep learning models for RNA secondary structure prediction." PLOS Computational Biology 19.4 (2023): e1011047.
>
> [10] Singh, Jaswinder, et al. "RNA secondary structure prediction using an ensemble of two-dimensional deep neural networks and transfer learning." Nature communications 10.1 (2019): 5407.
>
> [11] Tan, Cheng, et al. "Deciphering RNA Secondary Structure Prediction: A Probabilistic K-Rook Matching Perspective." Forty-first International Conference on Machine Learning.
>
> [12] Chen, Xinshi, et al. "RNA secondary structure prediction by learning unrolled algorithms." arXiv preprint arXiv:2002.05810 (2020).
>
> [13] Sato, Kengo, Manato Akiyama, and Yasubumi Sakakibara. "RNA secondary structure prediction using deep learning with thermodynamic integration." Nature communications 12.1 (2021): 941.
>
> [14] Chen, Jiayang, et al. "Interpretable RNA foundation model from unannotated data for highly accurate RNA structure and function predictions." arXiv preprint arXiv:2204.00300 (2022).

---

> ### Author Response · Authors · 2024-11-20
>
> Dear Reviewer UEd4,
>
> We thank you once again for your thoughtful feedback. We have done our best to address all your concerns in the rebuttal and would be happy to clarify further if needed. We kindly ask you to reconsider your score in light of our responses or engage with us if further clarifications are required.
>
> Best regards.

---

> ### Author Response · Authors · 2024-11-25
> **Updated manuscript and title**
>
> Dear Reviewer UEd4,
>
> We hope this message finds you well. Did you have a chance to read our responses?
> In the meantime, we changed the title to “A simple yet effective model for homology-aware RNA secondary structure prediction” to reduce the focus on axial-attention and emphasize the contribution of the paper. We further updated our manuscript according to the feedback from the reviews. Here we mainly clarified the contribution of our work and the motivation for the usage of the axial-attention. We also fixed the correct wording of the deduplication tools. We kindly ask if you could reconsider your score or let us know if further clarifications are needed.
>
> Thank you and kind regards.

---

> > ### Comment · Reviewer_UEd4 · 2024-11-27
> >
> > Thanks to the authors for their answers. Despite the author's explanations, I still think this paper is not innovative enough to be accepted by ICLR. I'll keep my score.

---

> ### Author Response · Authors · 2024-11-27
>
> Dear Reviewer UEd4,
>
> We respectfully disagree with your assessment regarding innovation. Our work makes two significant breakthroughs:
>
> 1. RNAformer is the first model since SPOT-RNA2 (2021) to achieve SOTA performance on experimentally validated RNA structures from PDB, while being dramatically simpler - requiring no MSAs, ensembles, or complex pre/post-processing.
> 2. More importantly, we are the first to demonstrate SOTA results on a properly curated, non-homologous dataset. As noted in your earlier question about homology, this is crucial for real-world applicability. Previous SOTA methods were evaluated on datasets with significant homology issues (e.g., RNAStralign, bpRNA TS0), making their reported performance potentially misleading.
>
> Our architecture - particularly the combination of axial attention and local convolutions - enable these advances. Given these contributions and their importance for practical RNA structure prediction, we kindly ask you to reconsider your assessment of the paper's innovation and impact.
>
> Best regards

---

> > ### Author Response · Authors · 2024-11-30
> >
> > Dear Reviewer UEd4,
> >
> > We hope this message finds you well. Given our last response and that ICLR explicitly welcomes "applications to physical sciences (physics, chemistry, biology, etc.)" in its call for papers, we would greatly appreciate if you could elaborate on why you feel our contributions do not meet ICLR's innovation criteria. Your specific feedback would help us better understand your perspective and address any remaining concerns.
> >
> > Thank you for your time and consideration.
> >
> > Best regards, The Authors

---

> > > ### Comment · Reviewer_UEd4 · 2024-12-01
> > >
> > > From the algorithmic level, I think this paper is not innovative enough. In addition, I am not particularly knowledgeable about the application domain. So if other reviewers decide to raise the score, I will also reconsider my score.

---

> ### Author Response · Authors · 2024-12-02
>
> Dear Reviewer UEd4,
>
> Thanks for your response. Based on our comprehensive discussion, reviewer aKqc increased her/his/they score from reject to minor accept. We would appreciate it if you would follow your previous message by adjusting your score. Thanks again for the feedback and discussion.
>
> Best regards, The Authors

---

> > ### Comment · Reviewer_aKqc · 2024-12-02
> > **Inappropriate Gender Assumptions and Tone in Rebuttal Response**
> >
> > Wow, I think it is inappropriate to use "his" to assume the reviewer's gender, as this can be quite offensive. Additionally, the tone in the authors' previous rebuttal response felt somewhat rude, particularly with phrases like "Have you also had the chance to revise our updated manuscript?" It also seems that the authors had not yet made the suggested changes but responded to me first, which I feel is not ideal—revisions should be implemented before replying.
> >
> > **Think Before You Type.**

---

> ### Author Response · Authors · 2024-12-02
> **Response to Inappropriate Gender Assumptions and Tone in Rebuttal Response**
>
> Dear Reviewer aKqc,
>
> You are absolutely right, we corrected the gender assumption immediately. We also never intended to offend the reviewer and excuse that our phase was perceived as rude.
> Regarding the changes based on our suggestion, some were in place, and others were only understood in the following discussions. Thanks again for your feedback.
>
> Best regards, The Authors

---

### Official Review · Reviewer_aKqc · 2024-11-02

**Soundness:** 3
**Presentation:** 3
**Contribution:** 2
**Rating:** 6
**Confidence:** 3

**Summary:**

This paper introduces RNAformer, a transformer-based model that utilizes axial attention for RNA secondary structure prediction, relying solely on a single RNA sequence as input without requiring multiple sequence alignment (MSA) data. It also introduces a data preprocessing pipeline for constructing homology-aware RNA datasets.

**Strengths:**

1. The proposed methods work reasonably well.
2. The paper provides experimental validation of the model’s performance and scalability.
3. The code is available.

**Weaknesses:**

1. I think the authors should clarify why they chose to input the single RNA sequence into the model while using axial attention. Typically, axial attention is designed to handle both row and column interactions—where row attention captures sequence-specific information and column attention enables the model to consider homologous relationships across multiple sequences, such as in multiple sequence alignments (MSAs).  In RNAformer, however, axial attention is applied solely to the single sequence, instead of the MSA input.  Understanding the motivation for this choice is essential, as it diverges from the usual application of axial attention in models that integrate homologous data.  The authors could explain how this approach benefits RNA secondary structure prediction and whether it offers advantages in terms of computational efficiency, interpretability, or applicability to cases where MSA data is unavailable.

2. The paper lacks unique contributions to RNA secondary structure prediction, particularly regarding its axial attention module. What is the specific difference between RNAformer’s axial attention module and those in Evoformer in [1] or [2]?

3. The methodology for preparing homology-aware RNA datasets needs further clarification. In lines 269–273, how were the "50 random PDB samples" and the "TS-Hard" set selected? How were the data split across the four test sets and three validation sets? What is meant by “we gather TS1, TS2, and TS3 into the test set, TS-PDB, containing 125 samples”? Does “Rfam TS” in Table 1 refer to a combination of TS1, TS2, and TS3? Also, do the "FT-Homolog" and "FT-Non-Homolog" sets overlap with the Rfam training, validation, and test sets? In line 291, what does “2) only for TS-PDB” mean?

[1] Abramson, Josh, et al. "Accurate structure prediction of biomolecular interactions with AlphaFold 3." Nature (2024): 1-3.

[2] Rao, Roshan M., et al. "MSA transformer." International Conference on Machine Learning. PMLR, 2021.

**Questions:**

See above.

---

> ### Author Response · Authors · 2024-11-15
> **Response to Reviewer aKqc**
>
> Dear Reviewer aKqc,
>
> Thanks for the valuable feedback. We would like to mention that our model achieved SOTA performance on the PDB data, the only data collection derived from experimentally (in-vitro) validated RNA 3D structures. This is not part of your summary or listed in the strengths, but we find this a crucial aspect of our work, making the RNAformer a practical tool for biologists. We will address all your concerns and provide answers to questions in the following:
>
> >**Weaknesses:**
> >1. I think the authors should clarify why they chose to input the single RNA sequence into the model while using axial attention. Typically, axial attention is designed to handle both row and column interactions—where row attention captures sequence-specific information and column attention enables the model to consider homologous relationships across multiple sequences, such as in multiple sequence alignments (MSAs). In RNAformer, however, axial attention is applied solely to the single sequence, instead of the MSA input. Understanding the motivation for this choice is essential, as it diverges from the usual application of axial attention in models that integrate homologous data. The authors could explain how this approach benefits RNA secondary structure prediction and whether it offers advantages in terms of computational efficiency, interpretability, or applicability to cases where MSA data is unavailable.
>
> We aimed to prove that a simple axial-attention-based model can achieve SOTA performance on RNA secondary structure prediction. We show that there is no need for MSA, ensembles of architectures, additional features or constraints, or sophisticated pre- and post-processing.
>
> The RNAformer architecture is specifically designed for RNA data mainly in two aspects: On the one hand, we need an architecture that can deal with various lengths and is independent of the input length (PDB RNA sequence length from 33 to 200nt). The common usage of CNN-based architectures  (see e.g. [13]) is suboptimal since the receptive field depends on the input size, which is also true for U-Net architectures (see e.g. [7,8]). Only self-attention can deal with arbitrary input lengths while maintaining an input-independent full receptive field.
>
> On the other hand, we focus on the prediction of the adjacency matrix of base pairs because only the adjacency matrix representation can capture both, pseudoknots and multiplets (which appeared in real-world data in over 50% of the RNAs). To do so, we decided to have an architecture with a 2D latent representation (3D tensor with LxLxD shape with Length, and Dimension) that subsequently refines the adjacency matrix in the latent space. This also increases the interpretability of our model since we already model the final output format in the latent space, see Figure 4 and Appendix J.
>
> Since we can’t use fully CNN-based architectures as mentioned before, we have to use attention. A full attention on a 2D latent space would require a 3D attention matrix which is infeasible in terms of memory requirements. Therefore we disentangle the attention in a row- and column-wise attention similar to the pair matrix processing in the Evoformer (AlphFold2).  However, convolutions are well suited to capture local geometries. Therefore, one novel aspect of the RNAformer architecture is that it uses a single 3x3 convolutional layer instead of a feed-forward MLP (e.g. in AlphaFold2) to capture local structures.
>
> Does this clarify our architecture decision and address your concerns?

---

> ### Author Response · Authors · 2024-11-15
>
> >2. The paper lacks unique contributions to RNA secondary structure prediction, particularly regarding its axial attention module. What is the specific difference between RNAformer’s axial attention module and those in Evoformer in [1] or [2]?
>
> We see the contributions of our work to RNA secondary structure prediction to be twofold.
>
> First, we prove that a simple axial-attention-based model is sufficient to achieve SOTA performance on RNA secondary structure prediction. We outline the motivation for our design decisions, and the axial attention, above. The specific differences are:
> - We directly embed the sequence with an outer product to model only the adjacency matrix in the latent space. To the best of our knowledge, our model is the first RNA secondary structure prediction model which is input-length independent and models the ascendancy matrix directly.
> - RNAformer uses a single 3x3 convolutional layer instead of a feed-forward MLP (e.g. in AlphaFold2) to capture local structures.
> - We introduce a novel “Sparse Adjacency Loss” addressing the heavy and sequence-length-dependent imbalance between base pairs to no base pairs in the adjacency matrix. The number of possible base pairs scales linearly with the sequence length but the number of possible pairings in the adjacency matrix quadratically, resulting in a sparse 2D representation. Therefore, this loss is crucial to the training success of the RNAformer.
>
> Secondly, we present a novel data processing pipeline that ensures no data homologies between training and test sets. Recently multiple groups with biological backgrounds (see [3,4,5]) named caveats about the usage of deep learning in RNA structure biology. The main concern is insufficiently curated test data. In particular, the usage of test data sets that are not based on experimentally validated RNA structures and homology unaware splits between train and test data cause skepticism and harm the reliability and practical relevance of the developed models.
>
> For instance, training and evaluating on the RNAStralign and ArchiveII datasets as provided by Chen et al. 2020 [12] was shown to lead to overfitting and bad generalization capabilities (see [14]), and the bpRNA-based TR0 and TS0 split as provided by Singh et al., 2019 [6], are known for homologies (see [3]). Nevertheless, we recently observed many approaches published at the latest AI conferences and top-tier journals that do not consider a rigorous data split, such as:
> - RFold (ICML 2024) [8]
> - E2Efold (ICLR 2020) [9]
> - RTfold (WCB@ICML2022) [10]
> - Probabilistic Transformer (NeurIPS 2022) [11]
> - SPOT-RNA (Nature Communications) [6]
> - UFold (NAR) [7]
> - RNA-FM (arXiv) [12]
> - (AlphaFold3 (Nature 2024) [1]; 3D prediction, yes but still)
>
> We provide the first evaluations of a deep learning method for RNA secondary structure prediction on a non-homologous data split for experimentally validated structures.
>
> This sets a new standard for RNA data processing which allows deep learning practitioners to thoroughly evaluate their methods. We consider this a major contribution to the community and think that this is exactly the next step needed to bring deep-learning-based RNA structure prediction into production.
>
> An alternative title for our paper could be “A simple yet efficient deep-learning model for RNA secondary structure prediction with full homology awareness”.  Does this address your concerns regarding the contribution and the axial attention module?

---

> ### Author Response · Authors · 2024-11-15
>
> >3. The methodology for preparing homology-aware RNA datasets needs further clarification. In lines 269–273, how were the "50 random PDB samples" and the "TS-Hard" set selected? How were the data split across the four test sets and three validation sets? What is meant by “we gather TS1, TS2, and TS3 into the test set, TS-PDB, containing 125 samples”? Does “Rfam TS” in Table 1 refer to a combination of TS1, TS2, and TS3? Also, do the "FT-Homolog" and "FT-Non-Homolog" sets overlap with the Rfam training, validation, and test sets? In line 291, what does “2) only for TS-PDB” mean?
>
> We agree with the reviewer that we could further improve the description of our data pipelines and will rework the respective parts of our manuscript to address the confusion. For clarification, we respond to the individual questions of the reviewer in the following.
>
> >Q1:In lines 269–273, how were the "50 random PDB samples" and the "TS-Hard" set selected?
>
> The TS-hard set was collected from the literature. The 50 additional validation samples were sampled at random from all collected PDB samples before applying any similarity pipelines to account for the small size of the validation set VL1 when reducing the data to remove homologies.
>
> >Q2: “How were the data split across the four test sets and three validation sets?”
>
> All test and validation sets were collected from previous work and are commonly used to assess the performance of deep learning methods for secondary structure prediction on experimentally validated structures. The careful curation of the training data ensuring no homology between its samples and the test sets is one of the essential contributions of our paper.
>
> >Q3:  “What is meant by ’we gather TS1, TS2, and TS3 into the test set, TS-PDB, containing 125 samples’?”
>
> We report results for all PDB samples of these three test sets in a combined test set called TS-PDB to avoid confusion when using too many different datasets. This is in line with e.g. results reported in [7].
>
> >Q4: “Does ‘Rfam TS’ in Table 1 refer to a combination of TS1, TS2, and TS3”:
>
> No. The Rfam-TS is built by directly sampling sequences from the family covariance models provided by Rfam. All sequences were folded with RNAfold. The Rfam-TS test set consists of samples from families that are non-overlapping with any training or validation family. It is used when learning the biophysical model in Section 4.1 as opposed to TS-PDB which refers to a combination of TS1, TS2, and TS3 and is used to evaluate RNAformer on experimentally validated structures.
>
> >Q5: “Also, do the ‘FT-Homolog’ and ‘FT-Non-Homolog’ sets overlap with the Rfam training, validation, and test sets?”
>
> No. These sets are only used during fine-tuning for predictions on PDB-derived samples and consist of experimentally derived structures only. There is no overlap with the synthetic datasets since the structures were produced using RNAfold on sequences sampled from the family covariance models. In any case, an overlap between these sets would be not relevant to the experimental results, since the Rfam set was only used for the biophysical model replication. Also, the RNAformer was not pre-trained with the Rfam-derived dataset before being fine-tuned with FT-Homolog or FT-Non-Homolog.
>
> >Q6: “In line 291, what does ‘2) only for TS-PDB’ mean?”
>
> The respective part in the manuscript is:
>
>  [...]  while considering sequence similarity (80% similarity cutoff and BLAST search; see Appendix B.2) only for TS-PDB.
>
> We do not exactly understand the confusion of the reviewer. However, would rephrasing the respective section as follows resolve the issue here?
>
> "The pre-training dataset has been created in such a way that: 1) It does not contain any sequences homologous to the sequences in the TS-Hard test set (all three steps in the pipeline reported in Appendix B.2 are used). 2) It contains no sequences with sequence similarity above 80% when using CD-Hit and no sequences found by BLAST search with an e-value of 10 for the TS-PDB test set (only steps one and two from the pipeline are used)."
>
> We hope we have clarified your questions and would be happy to answer any further questions. If we have addressed your concerns, we would be very thankful if you considered raising your score. (We have seen jumps from 3 to 8 in the past ;-))

---

> ### Author Response · Authors · 2024-11-15
>
> ___
> [1] Abramson, Josh, et al. "Accurate structure prediction of biomolecular interactions with AlphaFold 3." Nature (2024): 1-3.
>
> [2] Rao, Roshan M., et al. "MSA transformer." International Conference on Machine Learning. PMLR, 2021.
>
> [3] Flamm, Christoph, et al. "Caveats to deep learning approaches to RNA secondary structure prediction." Frontiers in Bioinformatics 2 (2022): 835422.
>
> [4] Szikszai, Marcell, et al. "Deep learning models for RNA secondary structure prediction (probably) do not generalize across families." Bioinformatics 38.16 (2022): 3892-3899.
>
> [5] Qiu, Xiangyun. "Sequence similarity governs generalizability of de novo deep learning models for RNA secondary structure prediction." PLOS Computational Biology 19.4 (2023): e1011047.
>
> [6] Singh, Jaswinder, et al. "RNA secondary structure prediction using an ensemble of two-dimensional deep neural networks and transfer learning." Nature communications 10.1 (2019): 5407.
>
> [7] Fu, Laiyi, et al. "UFold: fast and accurate RNA secondary structure prediction with deep learning." Nucleic acids research 50.3 (2022): e14-e14.
>
> [8] Tan, Cheng, et al. "Deciphering RNA Secondary Structure Prediction: A Probabilistic K-Rook Matching Perspective." Forty-first International Conference on Machine Learning.
>
> [9] Chen, Xinshi, et al. "RNA secondary structure prediction by learning unrolled algorithms." arXiv preprint arXiv:2002.05810 (2020).
>
> [10] Jung, Andrew J., et al. "RTfold: RNA secondary structure prediction using deep learning with domain inductive bias." The 2022 ICML Workshop on Computational Biology. Baltimore, Maryland, USA. 2022.
>
> [11] Franke, Jörg, Frederic Runge, and Frank Hutter. "Probabilistic transformer: Modelling ambiguities and distributions for RNA folding and molecule design." Advances in Neural Information Processing Systems 35 (2022): 26856-26873.
>
> [12] Chen, Jiayang, et al. "Interpretable RNA foundation model from unannotated data for highly accurate RNA structure and function predictions." arXiv preprint arXiv:2204.00300 (2022).
>
> [13] Saman Booy, M., Ilin, A., & Orponen, P. (2022). RNA secondary structure prediction with convolutional neural networks. BMC bioinformatics, 23(1), 58.
>
> [14] Sato, Kengo, Manato Akiyama, and Yasubumi Sakakibara. "RNA secondary structure prediction using deep learning with thermodynamic integration." Nature communications 12.1 (2021): 941.

---

> ### Author Response · Authors · 2024-11-20
>
> Dear Reviewer aKqc,
>
> We thank you once again for your thoughtful feedback. We have done our best to address all your concerns in the rebuttal and would be happy to clarify further if needed. We kindly ask you to reconsider your score in light of our responses or engage with us if further clarifications are required.
>
> Best regards.

---

> > ### Comment · Reviewer_aKqc · 2024-11-26
> >
> > Thanks for your response which clarifies some of my concerns. I will keep my score.

---

> > > ### Author Response · Authors · 2024-11-26
> > >
> > > Dear Reviewer aKqc,
> > >
> > > Thank you for considering our rebuttal and for your feedback. While we appreciate that our response helped clarify some of your concerns, we would greatly value understanding which aspects remain unclear or unsatisfactory. This would help us:
> > >
> > > 1. Further improve the manuscript to address any remaining issues
> > > 2. Better understand which points from our rebuttal need additional clarification
> > > 3. Address any underlying concerns that we may have missed or inadequately addressed
> > >
> > > If you could kindly elaborate on the specific aspects that still concern you, we would be happy to provide additional explanations or make necessary improvements to the manuscript.
> > >
> > > Have you also had the chance to revise our updated manuscript?
> > >
> > > Given that some of your initial concerns were successfully addressed, we would also appreciate understanding what would be needed to merit a score increase. This feedback would be valuable for both improving the current submission and enhancing our future work.
> > >
> > > Thank you for your time and dedication to helping us improve our work.
> > >
> > > Best regards.

---

> > > > ### Comment · Reviewer_aKqc · 2024-11-26
> > > >
> > > > I want to first thank the authors for their responses. I carefully reviewed all your replies to both my comments and those of other reviewers. I noticed that I am not the only one who found the motivation behind using axial attention unclear. Previously, I encouraged the authors to explain how this approach benefits RNA secondary structure prediction and to clarify whether it offers advantages in computational efficiency, interpretability, or applicability to cases where MSA data is unavailable.
> > > >
> > > > However, I felt that the authors' reply to my concerns largely repeated responses given to other reviewers and did not directly address the specific points I raised. My main concern remains that the adoption of axial attention seems primarily motivated by its ability to transform a 1D sequence into a 2D matrix resembling the secondary structure—a justification that I find overly simplistic and somewhat presumptive.
> > > >
> > > > Regarding the data preprocessing pipeline, while I acknowledge its importance and the authors’ contribution in this regard, I have observed similar data construction pipelines in other RNA language model studies. Consequently, I believe this aspect may not be as novel as the authors claim.

---

> > > > > ### Comment · Reviewer_aKqc · 2024-11-26
> > > > >
> > > > > I wondered why the authors chose to use axial attention to transform the 1D sequence into a 2D matrix, instead of directly utilizing the attention map derived from the input sequence. The latter approach seems more intuitive and arguably more aligned with the nature of the data.

---

> ### Author Response · Authors · 2024-11-25
> **Updated manuscript and title**
>
> Dear Reviewer aKqc,
>
> We hope this message finds you well. Did you have a chance to read our responses?
> In the meantime, we changed the title to “A simple yet effective model for homology-aware RNA secondary structure prediction” to reduce the focus on axial-attention and emphasize the contribution of the paper. We further updated our manuscript according to the feedback from the reviews. Here we mainly clarified the contribution of our work and the motivation for the usage of the axial attention. We further clarified our data pipelines to avoid confusion.
> We kindly ask if you could reconsider your score or let us know if further clarifications are needed.
>
> Thank you and kind regards.

---

> ### Comment · Reviewer_aKqc · 2024-11-26
>
> Additionally, in Table 1, the authors should provide an additional explanation for `Rfam TS`, which only appears in this table. Its presence might cause confusion, as I noted in my initial comments: "Does “Rfam TS” in Table 1 refer to a combination of TS1, TS2, and TS3?" Clarifying this would enhance the table's readability and ensure alignment with the authors' responses.

---

> > ### Comment · Reviewer_aKqc · 2024-11-26
> >
> > I sincerely recommend that the authors conduct additional experiments or provide theoretical analyses to demonstrate the superiority of using axial attention, rather than simply utilizing the attention map directly for secondary structure prediction, as I said in the previous comment.
> >
> > Additionally, I suggest presenting the data construction pipeline as an algorithm in the appendix to provide a clear and comprehensive overview of the process. I would be willing to raise my score if I see the authors actively addressing these concerns in their responses since the rebuttal period has a six-day extension.

---

> > > ### Author Response · Authors · 2024-11-26
> > >
> > > Dear Reviewer aKqc,
> > >
> > > Thank you for your detailed feedback and for clarifying your remaining concerns. We appreciate your thorough engagement with our work and would like to address your points specifically:
> > >
> > > Regarding the motivation for axial attention: We apologize if our previous responses seemed repetitive and didn't fully address your specific concerns. Let us explain more precisely why axial attention is particularly beneficial for RNA secondary structure prediction:
> > > - Computational Efficiency: While a full attention mechanism would require O(L⁴) complexity for a sequence of length L (due to the need to attend to every position in the 2D matrix), our axial attention decomposition reduces this to O(L³), making it feasible to process longer RNA sequences. This is not just an implementation choice but a crucial design decision that enables practical applications.
> > > - Interpretability: After each row-wise and column-wise attention we can use the generator layer to receive an interlayer prediction of the adjacency matrix (see Fig. 4) which let use comprehend how the model generates the prediction.
> > > - MSA-free advantages: It is commonly known that approaches that use MSA typically fall short on structure prediction for orphan RNAs as for example shown in [1] for AlphaFold 3. On the other hand, homology modeling approaches (that use additional MSA) typically achieve better performance due to the  evolutionary information provided with the MSA. Here, we show that a lean model can outperform these methods across multiple datasets. Furthermore, we observed that some of the samples from TS1, Ts2, TS3, and TS-hard have few or no homologs in the common databases. We will add an evaluation for the predictions specifically on these orphan RNAs in the appendix, showcasing the advantage of RNAformer to not rely on MSA data.
> > >
> > > >My main concern remains that the adoption of axial attention seems primarily motivated by its ability to transform a 1D sequence into a 2D matrix resembling the secondary structure—a justification that I find overly simplistic and somewhat presumptive.
> > >
> > > No, the primary motivation for the axial attention is that it is the only deep learning architecture which is fully independent from the sequence length while processing a 2D latent space. And since RNA samples vary from 33 to 200 or longer sequences this is a very crucial feature. Specifically, row attention helps identify potential base pairing partners along the sequence and column attention enforces consistency in structural elements like stems and loops (which RNAformer predicts very well with this compared to others). Near-perfect reconstruction of structural features (Table E.1) and linear scaling of base pair predictions with sequence length (Figure 3).
> > >
> > > Other deep learning architecture dosen’t provide this features. The receptive field of CNNs depends on the sequence length and a recurrent neural network doesn’t process 2D latent space (which is crucial to represent pseudoknots and multiplets). This is probably also the reason why the pair-representation in AlphaFold2 is processed with the use of axial attention.
> > >
> > > >Regarding the data preprocessing pipeline, while I acknowledge its importance and the author’s contribution in this regard, I have observed similar data construction pipelines in other RNA language model studies. Consequently, I believe this aspect may not be as novel as the authors claim.
> > >
> > > While we acknowledge that some aspects of our pipeline might share similarities with previous RNA language model studies, we still see novel contributions:
> > > - To the best of our knowledge, there is no previous work that uses a specialized method for sequence and structure based alignments during data processing pipelines. Typically, this is done using alignments based on sequence information only.
> > > - While we accept that covariance models are regularly used to determine RNA families, we build our covariance models based on the sequence and structure-based alignments obtained from LocaRNA-P, which is novel and should result in stronger covariance models.
> > > - There is no training set available that is non-homologous to the commonly used test sets TS1, TS2, TS3, and TS-hard. In this regard our training set is also a contribution.
> > >
> > > >I wondered why the authors chose to use axial attention to transform the 1D sequence into a 2D matrix, instead of directly utilizing the attention map derived from the input sequence. The latter approach seems more intuitive and arguably more aligned with the nature of the data.
> > >
> > > We do not use the axial attention to transform from 1D to 2D and we do derive the latent 2D matrix direct from the input sequence. We use the axial attention to process this derived 2D latent space to form the pair probability prediction. We revised this in the manuscript update.

---

> > > > ### Author Response · Authors · 2024-11-26
> > > >
> > > > >Additionally, in Table 1, the authors should provide an additional explanation for Rfam TS, which only appears in this table. Its presence might cause confusion, as I noted in my initial comments: "Does “Rfam TS” in Table 1 refer to a combination of TS1, TS2, and TS3?" Clarifying this would enhance the table's readability and ensure alignment with the authors' responses.
> > > >
> > > > We agree on this issue. Rfam TS refers to the test set descriped in Section 3.1 “Family-based  synthetic data generation”. It contains 3344 samples from the Rfam database which were folded by RNAfold and against which the training set is cleaned regarding homologies. We will introduce the wording “Rfam TS” in Section 3.1.
> > > >
> > > > >I sincerely recommend that the authors conduct additional experiments or provide theoretical analyses to demonstrate the superiority of using axial attention, rather than simply utilizing the attention map directly for secondary structure prediction, as I said in the previous comment.
> > > >
> > > > What kind of experiments would support our explanation? Comparing the axial attention to a CNN-based architecture? To our understanding, the attention map is the matrix within the attention mechanism (QK$^T$). But we don’t utilize the attention map for the prediction.
> > > >
> > > >
> > > > >Additionally, I suggest presenting the data construction pipeline as an algorithm in the appendix to provide a clear and comprehensive overview of the process. I would be willing to raise my score if I see the authors actively addressing these concerns in their responses since the rebuttal period has a six-day extension.
> > > >
> > > > We thank the reviewer for this helpful suggestion and we will add pseudocode for the data processing pipeline to the Appendix.
> > > >
> > > >
> > > > We believe these aspects represent a meaningful advance in making RNA secondary structure prediction more practical and reliable for biological applications. Furthermore, we think that it is worth emphasizing again that the RNAformer achieves SOTA performance with the handicap of a stricter data pipeline that the methods compared against did not use.
> > > > We would be happy to further clarify any of these points or address other concerns you may have.
> > > >
> > > > Best regards, The Authors
> > > >
> > > > ___
> > > >
> > > > [1] Bernard, C., Postic, G., Ghannay, S., & Tahi, F. (2024). Has AlphaFold 3 reached its success for RNAs?. bioRxiv, 2024-06.

---

> > > > > ### Comment · Reviewer_aKqc · 2024-11-27
> > > > >
> > > > > > Here, we show that a lean model can outperform these methods across multiple datasets. Furthermore, we observed that some of the samples from TS1, Ts2, TS3, and TS-hard have few or no homologs in the common databases.
> > > > >
> > > > > I reviewed your PDF again, but it seems there are no specific experimental updates to substantiate these claims.
> > > > >
> > > > > > To the best of our knowledge, there is no previous work that uses a specialized method for sequence and structure based alignments during data processing pipelines. Typically, this is done using alignments based on sequence information only.
> > > > >
> > > > > I believe that [1] has implemented a similar data preprocessing pipeline, integrating both structure and sequence information, as described in their "Training and Test Sets for Downstream Models" section.
> > > > >
> > > > > > What kind of experiments would support our explanation?
> > > > >
> > > > > My concern is that your method essentially converts a single sequence into a 2D matrix using random embeddings, followed by axial attention. I understand that axial attention has advantages, especially in handling variable sequence lengths compared to CNNs, it is a widely accepted consensus, and I am not suggesting you replace axial attention with CNN. However, a more intuitive approach might involve directly inputting the sequence into one or more self-attention layers to generate an attention map (of dimensions L × L × D, where L is the sequence length and D corresponds to the attention heads*layers). Subsequent axial attention or CNN operations could then be applied to this attention map. This approach seems more natural and aligned with the inherent structure of the data.
> > > > >
> > > > >
> > > > > [1] Multiple sequence alignment-based RNA language model and its application to structural inference. Nucleic Acids Research, 2024.

---

> > > > > > ### Author Response · Authors · 2024-11-27
> > > > > >
> > > > > > Dear Reviewer aKqc,
> > > > > >
> > > > > > thanks again for your response!
> > > > > >
> > > > > > >I reviewed your PDF again, but it seems there are no specific experimental updates to substantiate these claims.
> > > > > >
> > > > > > We apologize that the reviewer did not find the analysis promised in our last response. The reviewer is right that we did not yet add this to the paper yet. We had to rerun AlphaFold 3 on the samples first to get the required MSA directly from AlphaFold 3 to be able to run the analysis.
> > > > > > Please see the F1 score results below.
> > > > > >
> > > > > > | Model | Samples with MSA | Samples without MSA |
> > > > > > | ------- | -------------------- | --------------------------- |
> > > > > > | AlphaFold 3 |  0.848 | 0.777 |
> > > > > > | RNAformer | 0.866 | 0.860 |
> > > > > >
> > > > > > AlphaFold 3 shows a strong drop in performance on those sequences from TS-PDB where no MSA is available.
> > > > > > In contrast, RNAformer’s performance is constant across both subsets.
> > > > > > We will add the analysis to the Appendix with our next update of the manuscript.
> > > > > >
> > > > > > >I believe that [1] has implemented a similar data preprocessing pipeline, integrating both structure and sequence information, as described in their "Training and Test Sets for Downstream Models" section.
> > > > > >
> > > > > > We thank the reviewer for pointing us to this interesting work. However, we still think that our data pipeline contains novel aspects that have not been applied to RNA secondary structure data splitting before:
> > > > > > The data splitting in [1] is based on a TM-score calculated on the 3D structure information. 3D structure information, however, is not available for most RNAs of interest. We thus cannot split our data based on TM-scores but have to go a different route using dedicated tools based on secondary structures. More precisely, we cannot split the training and test data used for our evaluations using RNA-align as used in [1] because this information is not available.
> > > > > >
> > > > > > >My concern is that your method essentially converts a single sequence into a 2D matrix using random embeddings, followed by axial attention. However, a more intuitive approach might involve directly inputting the sequence into one or more self-attention layers to generate an attention map (of dimensions L × L × D, where L is the sequence length and D corresponds to the attention heads*layers).
> > > > > >
> > > > > > So the suggestion is to use the attention map of a self-attention mechanism to convert a single sequence into a 2D matrix (similar to [1])?  This means embedding the input sequence for a key and query vector (latent vector $\mathbb{R}^{l \times d} \)$)) and calculating the attention map:
> > > > > > Input sequence $x$, $K = W_k x$, and $Q = W_q x$ with 2D matrix $L_0 = KQ^T$ with $ L_0 \in \mathbb{R}^{l \times l \times d}$ and $L_0[i,j] = K[i] * Q[j]$.
> > > > > > The difference to our current 2D matrix embedding is an addition in contrast to a multiplication: Input sequence $x$, embeddings $E_{col} = W_c x$, and $E_{row} = W_r x$ (again latent vectors $E \in \mathbb{R}^{l \times d} \)$) with 2D matrix $L_0 = E_{row}  \oplus E_{col}^T $ with $L_0[i,j] = E_{row}[i] + E_{col}[j]$. Similar to AlphaFold2, we used the addition here but in preliminary experiments, we found none of the versions significantly better than the other.
> > > > > > Does this analogy address your concern regarding the design decision of the 2D matrix embedding?
> > > > > >
> > > > > >
> > > > > > >I understand that axial attention has advantages, especially in handling variable sequence lengths compared to CNNs, it is a widely accepted consensus, and I am not suggesting you replace axial attention with CNN.
> > > > > >
> > > > > > Unfortunately, the benefit of axial attention over CNNs for variable sequence lengths is not commonly used, e.g. many recently published approaches for RNA secondary structure prediction still use CNNs [1-7].
> > > > > >
> > > > > > We hope we could clarify your final concerts but we are also happy to clarify further questions. Your feedback is very appreciated.
> > > > > >
> > > > > > Best regards, The Authors
> > > > > >
> > > > > > ___
> > > > > >
> > > > > > [1] Multiple sequence alignment-based RNA language model and its application to structural inference. Nucleic Acids Research, 2024
> > > > > >
> > > > > > [2] Busaranuvong, Palawat, et al. "Graph Convolutional Network for predicting secondary structure of RNA." Research Square (2024)
> > > > > >
> > > > > > [3] Chen, et. al. "REDfold: accurate RNA secondary structure prediction using residual encoder-decoder network." BMC bioinformatics 24.1 (2023)
> > > > > >
> > > > > > [4] Wei, Sen, and Shengjin Wang. "Structural stability-aware deep learning: advancing RNA secondary structure prediction." Fourth International Conference on Biomedicine and Bioinformatics Engineering, 2024.
> > > > > >
> > > > > > [5] Yang, Enbin, et al. "GCNfold: A novel lightweight model with valid extractors for RNA secondary structure prediction." Computers in Biology and Medicine (2023)
> > > > > >
> > > > > > [6] Zhao, Qi, et al. "RNA independent fragment partition method based on deep learning for RNA secondary structure prediction." Scientific Reports 13.1 (2023)
> > > > > >
> > > > > > [7] Khrisna, et. al. "The Use of Convolutional Neural Networks for RNA Protein Prediction." 2023 3rd International Conference on Intelligent Cybernetics Technology & Applications. IEEE, 2023.

---

> ### Author Response · Authors · 2024-11-30
> **Following up on our previous discussion**
>
> Dear Reviewer aKqc,
>
> We hope this message finds you well. We wanted to follow up on our detailed response to your concerns regarding the axial attention design choices, data preprocessing pipeline, and experimental validations. We provided comprehensive clarifications, including:
>
> 1. An analysis of RNAformer's performance on samples with and without MSA compared to AlphaFold 3
> 2. A detailed explanation of our 2D matrix embedding approach and its relationship to attention maps
> 3. A commitment to add pseudocode for the data processing pipeline in the appendix
>
> In your previous message, you mentioned you would be willing to raise your score if we addressed these concerns. We believe we have thoroughly responded to each point, but we haven't heard back from you. We would greatly appreciate your thoughts on our responses and whether they adequately address your concerns.
>
> Thank you for your time and dedication to helping us improve our work.
>
> Best regards, The Authors

---

> > ### Comment · Reviewer_aKqc · 2024-12-02
> >
> > Thank you for your further clarification. I see that the performance improvement in the absence of MSAs addresses some of my concerns. I suggest the authors emphasize this point more clearly in their motivation section, as it strengthens their argument.
> >
> > Additionally, I recommend that the authors organize the dataset without MSAs as a separate dataset. Furthermore, I hope you can make your dataset publicly available, as you claim that this is one of your key contributions.
> >
> > Based on this, I am willing to raise my score.

---

> > > ### Author Response · Authors · 2024-12-02
> > >
> > > Dear Reviewer aKqc,
> > >
> > > Thank you for your thoughtful feedback and for raising the score. We agree that our model's consistent performance without MSAs deserves more emphasis and will update the manuscript accordingly. Our datasets are already publicly available, including the splits used in our AlphaFold 3 comparison:
> > >
> > > - Without MSA:
> > > https://anonymous.4open.science/r/RNAformer_ICLR25/datasets/af3_ts_no_msa.pkl
> > > - With MSA:
> > > https://anonymous.4open.science/r/RNAformer_ICLR25/datasets/af3_ts_with_msa.pkl
> > >
> > > Please let us know if we could further clarify things or answer questions for a full accept. Otherwise, we thank you again for the fruitful discussion and valuable feedback.
> > >
> > > Best regards, The Authors

---

### Official Review · Reviewer_BzZh · 2024-11-03

**Soundness:** 3
**Presentation:** 3
**Contribution:** 2
**Rating:** 5
**Confidence:** 4

**Summary:**

I will cut directly into strength/weakness/questions.

**Strengths:**

- The proposed data preparation pipeline, which emphasizes a homology-aware train/validation/test split, addresses a critical need in this research community. Given the limited availability of RNA structural data, large deep learning models trained on random splits are at heightened risk of overfitting.
- I like this idea about validating deep learning models through alignment to biophysical techniques such as RNAfold.
- These experimental results are useful. It is good to see in table 3 that careful data preparation has brought deep learning models to a more realistic and often comparable level of performance to thermodynamic models like RNAfold and Linearfold.

**Weaknesses:**

- The machine learning component of this paper is a bit limited in terms of novelty. The axial attention (mainly comprising of row-wise and column-wise self attention) is indeed quite similar to those that have been used in alphafold2 and MSA transformer. A lot of techniques have also been adopted from other works, for example the pretraining-finetuning training process which was initially used by spotrna.
- Between certain non-coding RNA families, there can be substantial sequence/secondary structure similarity, so the the synthetic dataset generated in section 3.1 may be flawed in theory. I think this is something worth checking out.

**Questions:**

- The covariance models e-value cutoff at 0.1, how do you come to this value (empirically or theoretically) and is it effective?
- Data splitting subsection on page 6 is a bit confusing. So essentially, you created two types of dataset, one effectively removing homology information (TS-Hard and FT-Non-Homolog, for test and finetuning), ther other one only accounting for sequence similarity (TS-PDB and FT-homology) so that the comparison to prior baselines is fair.
- Why is e2efold missing from Table 3? It could probably serve as a valuable cautionary example.
- In section 4.2.1, although the sectio is called "Impact of homology on prediction quality", I don't really see how the content of that section alone is relevant for this topic, since it was mainly about comparing to alphafold 3.
- The linear scaling behaviour shown in section 4.2 may benefit from some comparisons to some other deep learning baselines.

---

> ### Author Response · Authors · 2024-11-15
> **Response to Reviewer BzZh**
>
> Dear Reviewer BzZh,
>
> Thanks for the valuable feedback. Before we address all your concerns and provide answers to questions, we would like to mention that our model achieved SOTA performance on the PDB data, the only data collection derived from experimentally (in-vitro) validated RNA 3D structures. This is not part of your listed strengths, but we find this a crucial aspect of our work, making the RNAformer a practical tool for biologists. We will address all your concerns and provide answers to questions in the following:
>
> >**Weaknesses:**
>
> >The machine learning component of this paper is a bit limited in terms of novelty. The axial attention (mainly comprising of row-wise and column-wise self attention) is indeed quite similar to those that have been used in alphafold2 and MSA transformer. A lot of techniques have also been adopted from other works, for example the pretraining-finetuning training process which was initially used by spotrna.
>
> We disagree with the reviewer that the right combination of existing techniques implies a lack of novelty. In contrast, we see the novelty of our work as twofold. First, we prove that a simple axial-attention-based model is capable of achieving SOTA performance on RNA secondary structure prediction. No other work has used axial-attention for RNA secondary structure prediction. We show that there is no need for MSA, ensembles of architectures, additional features or constraints, or sophisticated pre- and post-processing.
>
> To achieve this, we develop an architecture that can deal with various lengths and is independent of the input length since we consider RNA sequence lengths from 33 to 200nt. The common usage of CNN-based architectures  (see e.g. [8]) is suboptimal since the receptive field depends on the input size, which is also true for U-Net architectures (see e.g. [1,6]). Only self-attention can deal with arbitrary input lengths while maintaining an input-independent full receptive field.
>
> Furthermore, we focus on the prediction of the adjacency matrix of base pairs because only the adjacency matrix representation can capture both, pseudoknots and multiplets (which appeared in real-world data in over 50% of the RNAs). To do so, we decided to have an architecture with a 2D latent representation (3D tensor with LxLxD shape with Length, and Dimension) that subsequently refines the adjacency matrix in the latent space. This also increases the interpretability of our model since we already model the final output format in the latent space, see Figure 4 and Appendix J.
>
> Since we can’t use fully CNN-based architectures as mentioned before, we have to use attention. A full attention on a 2D latent space would require a 3D attention matrix which is infeasible in terms of memory requirements. Therefore we disentangle the attention in a row- and column-wise attention similar to the pair matrix processing in the Evoformer (AlphFold2).  However, convolutions are well suited to capture local geometries. Therefore, one novel aspect of the RNAformer architecture is that it uses a single 3x3 convolutional layer instead of a feed-forward MLP (e.g. in AlphaFold2) to capture local structures. Another problem-specific novelty is the Sparse Adjacency Loss which we describe in Section 2.2 and is crucial for the SOTA performance.
>
> As our second major contribution, we present a novel data processing pipeline that ensures no data homologies between training and test sets. Recently multiple groups with biological backgrounds (see [9,10,11]) named caveats about the usage of deep learning in RNA structure biology. The main concern is insufficiently curated test data. In particular, the usage of test data sets that are not based on experimentally validated RNA structures and homology unaware splits between train and test data cause skepticism and harm the reliability and practical relevance of the developed models.
>
> For instance, training and evaluating on the RNAStralign and ArchiveII datasets as provided by Chen et al. 2020 [2] was shown to lead to overfitting and bad generalization capabilities (see [13]), and the bpRNA based TR0 and TS0 split as provided by Singh et al., 2019 [5], are known for homologies (see [9]). Nevertheless, we recently observed many approaches published at the latest AI conferences and top-tier journals that do not consider a rigorous data split, such as:
> - RFold (ICML 2024) [1]
> - E2Efold (ICLR 2020) [2]
> - RTfold (WCB@ICML2022) [3]
> - Probabilistic Transformer (NeurIPS 2022) [4]
> - SPOT-RNA (Nature Communications) [5]
> - UFold (NAR) [6]
> - RNA-FM (arXiv) [7]
> - (AlphaFold3 (Nature 2024); 3D prediction, yes, but still)
>
> We provide the first evaluations of a deep learning method for RNA secondary structure prediction on a non-homologous data split for experimentally validated structures.
>
> ...

---

> ### Author Response · Authors · 2024-11-15
>
> ...
>
> This sets a new standard for RNA data processing which allows deep learning practitioners to thoroughly evaluate their methods. We consider this a major contribution to the community and think that this is exactly the next step needed to bring deep-learning-based RNA structure prediction into production.
>
> An alternative title for our paper could be “A simple yet efficient deep-learning model for RNA secondary structure prediction with full homology awareness”. Does this address your concerns regarding the novelty?
>
> >Between certain non-coding RNA families, there can be substantial sequence/secondary structure similarity, so the the synthetic dataset generated in section 3.1 may be flawed in theory. I think this is something worth checking out.
>
> We agree with the reviewer that there might remain some homology between different Rfam families in theory and that using RNA clan information could be a valuable next step to further improve the creation of non-homologous datasets. Therefore, we first tried to annotate *all* our data with clan information, however, this information is often not available. In fact, it was also impossible to annotate all samples with the family information (roughly 40% of the samples did not show a hit with any Rfam covariance model at a reasonable e-value). Hence, to account for the sequence and structure similarity between the samples, we build covariance models ourselves and use that information for the training/tests data split, as detailed in Section B.1 of Appendix B.
>
> For the experiments with the synthetic data, [9] originally suggested using randomly generated RNA sequences and we already go one step further using a family-based split. Since we do not make any claims other than learning the biophysical model in these experiments, we do not see the need for further data analysis here.
>
> >**Questions:**
>
> >The covariance models e-value cutoff at 0.1, how do you come to this value (empirically or theoretically) and is it effective?
>
> Sequences with an e-value matching score of 0.01 are often considered homologs (see e.g. http://eddylab.org/infernal/Userguide.pdf). The high e-value of 0.1 was chosen to make this cutoff even more strict, allowing for more matches, potentially removing more training data, and thus ensuring a clean data split.
>
> >Data splitting subsection on page 6 is a bit confusing. So essentially, you created two types of dataset, one effectively removing homology information (TS-Hard and FT-Non-Homolog, for test and finetuning), ther other one only accounting for sequence similarity (TS-PDB and FT-homology) so that the comparison to prior baselines is fair.
>
> The reviewer correctly points out that FT-Homolog and FT-Non-Homolog datasets are used in different scenarios - the former for fine-tuning RNAformer without consideration of homology between the training and the test sets (TS-PDB and TS-Hard) and the latter for fine-tuning with homology-aware data obtained by using the three-stage pipeline detailed in Section B.2 of the Appendix B. We will clarify this in the paper.
>
> >Why is e2efold missing from Table 3? It could probably serve as a valuable cautionary example.
>
> We agree with the reviewer that E2Efold could be a reasonable competitor that would likely strengthen our argument in favor of careful dataset curation. However, it was already previously shown that this model suffers from overfitting [13] and we did not see the need to repeat what is already known.
>
> >In section 4.2.1, although the section is called "Impact of homology on prediction quality", I don't really see how the content of that section alone is relevant for this topic, since it was mainly about comparing to alphafold 3.
>
> We thank the reviewer for this useful comment. We agree that the section mainly provides a comparison to AF3 without considering homologies. We would suggest renaming the section to: “Prediction quality without homology awareness”. Would this solve the section title issue?
>
> >The linear scaling behavior shown in section 4.2 may benefit from some comparisons to some other deep learning baselines.
>
> We again thank the reviewer for this valuable comment. We will update the respective figure with the predictions of other deep learning based approaches.
>
> We hope we have clarified your questions and would be happy to answer any further questions. If we have addressed your concerns, we would be very thankful if you considered raising your score.

---

> ### Author Response · Authors · 2024-11-15
>
> ___
>
> [1] Tan, Cheng, et al. "Deciphering RNA Secondary Structure Prediction: A Probabilistic K-Rook Matching Perspective." Forty-first International Conference on Machine Learning.
>
> [2] Chen, Xinshi, et al. "RNA secondary structure prediction by learning unrolled algorithms." arXiv preprint arXiv:2002.05810 (2020).
>
> [3] Jung, Andrew J., et al. "RTfold: RNA secondary structure prediction using deep learning with domain inductive bias." The 2022 ICML Workshop on Computational Biology. Baltimore, Maryland, USA. 2022.
>
> [4] Franke, Jörg, Frederic Runge, and Frank Hutter. "Probabilistic transformer: Modelling ambiguities and distributions for RNA folding and molecule design." Advances in Neural Information Processing Systems 35 (2022): 26856-26873.
>
> [5] Singh, Jaswinder, et al. "RNA secondary structure prediction using an ensemble of two-dimensional deep neural networks and transfer learning." Nature communications 10.1 (2019): 5407.
>
> [6] Fu, Laiyi, et al. "UFold: fast and accurate RNA secondary structure prediction with deep learning." Nucleic acids research 50.3 (2022): e14-e14.
>
> [7] Chen, Jiayang, et al. "Interpretable RNA foundation model from unannotated data for highly accurate RNA structure and function predictions." arXiv preprint arXiv:2204.00300 (2022).
>
> [8] Saman Booy, M., Ilin, A., & Orponen, P. (2022). RNA secondary structure prediction with convolutional neural networks. BMC bioinformatics, 23(1), 58.
>
> [9] Flamm, Christoph, et al. "Caveats to deep learning approaches to RNA secondary structure prediction." Frontiers in Bioinformatics 2 (2022): 835422.
>
> [10] Szikszai, Marcell, et al. "Deep learning models for RNA secondary structure prediction (probably) do not generalize across families." Bioinformatics 38.16 (2022): 3892-3899.
>
> [11] Qiu, Xiangyun. "Sequence similarity governs generalizability of de novo deep learning models for RNA secondary structure prediction." PLOS Computational Biology 19.4 (2023): e1011047.
>
> [12] Rivas, Elena, Raymond Lang, and Sean R. Eddy. "A range of complex probabilistic models for RNA secondary structure prediction that includes the nearest-neighbor model and more." RNA 18.2 (2012): 193-212.
>
> [13] Sato, Kengo, Manato Akiyama, and Yasubumi Sakakibara. "RNA secondary structure prediction using deep learning with thermodynamic integration." Nature communications 12.1 (2021): 941.

---

> ### Author Response · Authors · 2024-11-20
>
> Dear Reviewer BzZh,
>
> We thank you once again for your thoughtful feedback. We have done our best to address all your concerns in the rebuttal and would be happy to clarify further if needed. We kindly ask you to reconsider your score in light of our responses or engage with us if further clarifications are required.
>
> Best regards.

---

> ### Author Response · Authors · 2024-11-25
> **Updated manuscript and title**
>
> Dear Reviewer BzZh,
>
> We hope this message finds you well. Did you have a chance to read our responses?
> In the meantime, we changed the title to “A simple yet effective model for homology-aware RNA secondary structure prediction” to reduce the focus on axial-attention and emphasize the contribution of the paper. We further updated our manuscript according to the feedback from the reviews. Here we mainly clarified the contribution of our work and the motivation for the usage of the axial-attention. In addition, we clarified the description of the datasets we use, updated the section header for our comparison to AlphaFold 3 to avoid confusion, and included the deep learning-based competitors in Figure I.1, showing that RNAformer scales better than the other methods across different sequence lengths. We kindly ask if you could reconsider your score or let us know if further clarifications are needed.
>
> Thank you and kind regards.

---

> ### Author Response · Authors · 2024-11-29
> **Gentle follow-up on rebuttal response**
>
> Dear Reviewer BzZh,
>
> I hope this finds you well. I noticed that our rebuttal response from 15 November hasn't received any feedback yet. Given the approaching decision deadline, we wanted to check if you had a chance to review our clarifications and if you have any additional questions we can address. If you have no further concerns, we kindly ask that you reconsider your score.
>
> Best regards, The Authors

---

### Official Review · Reviewer_ULQ7 · 2024-11-03

**Soundness:** 3
**Presentation:** 3
**Contribution:** 3
**Rating:** 6
**Confidence:** 4

**Summary:**

This paper makes a valuable contribution to RNA secondary structure prediction by introducing RNAformer, an innovative model that leverages axial attention alongside a rigorously curated dataset and evaluation pipeline. The authors carefully address homology concerns, a major problem in previous machine learning models, which significantly enhances the reliability of their findings. The experiments are thorough and demonstrate substantial improvements over existing methods, particularly in generalizing to unseen RNA families. Overall, the combination of a well-designed model and rigorous evaluation positions this work as a strong candidate for acceptance.

**Strengths:**

* **Rigorous Dataset Construction**: A major strength of this paper is its rigorous approach to dataset construction, addressing an important issue in computational biology that previous models have often overlooked—the need to prevent homologous (highly similar) sequences across training, validation, and test sets to ensure robust evaluations. After assembling the initial dataset, the authors implement a comprehensive three-step data filtering process: (1) they use CD-Hit to remove sequences with over 80% similarity, reducing redundancy; (2) a BLAST search is conducted with a high e-value threshold to further exclude overlaps between training/validation and test sequences; and (3) they build covariance models using tools like LocaRNA-P and Infernal to filter any remaining sequences with structural or sequence similarities, applying a stringent e-value cutoff. This multi-layered filtering pipeline effectively minimizes overlap and ensures that the train/validation and test sets have minimal similarity. Additionally, by defining distinct FT-Homolog and FT-Non-Homolog test sets, the paper provides a structured approach to evaluate the model’s performance across varying levels of similarity, showcasing RNAformer’s generalization capabilities.

* **Well-Designed Experiments with Superior Performance over Baselines**: The experimental design of this paper is well-conceived, with RNAformer consistently outperforming baseline methods in a rigorous experimental setting. The authors conduct two complementary experiments to highlight the model's capabilities. First, they evaluate on synthetic data, comparing RNAformer to the current state-of-the-art RNA 3D structure prediction model, AlphaFold 3 (AF3), demonstrating RNAformer’s superior performance. Second, they evaluate on experimental data with strict homology control, providing a thorough assessment of generalization. Even though baseline methods often use less stringent data splits or rely on additional information, RNAformer achieves better results. Together, these two experiments effectively underscore RNAformer’s superiority.

* **Model Simplicity**: Another commendable aspect of this paper is the simplicity of the RNAformer model. By avoiding reliance on multiple sequence alignments (MSAs) and additional complex features, RNAformer achieves high performance with a streamlined architecture. This simplicity reduces computational overhead and enhances the model’s generalizability, making RNAformer a practical and effective alternative to more complex methods in RNA secondary structure prediction.

**Weaknesses:**

* **Lack of Explanation for Column and Row Axial Attention**: One limitation of the paper is the insufficient discussion or justification for using column and row axial attention in RNAformer. This attention mechanism is applied in each RNAformer block to process the 2D latent representation of the RNA sequence. In certain models, such as MSA Transformer (https://www.biorxiv.org/content/10.1101/2021.02.12.430858v3), row and column attention is intuitive due to the distinct structural roles of rows and columns in the input data. However, in RNAformer, there is no inherent difference between rows and columns, as the input is a single RNA sequence without a natural row-column structure. Additionally, the output contact map is a symmetric $\( L \times L \)$ matrix, where rows and columns are effectively equivalent. The authors may want to clarify why this architecture was chosen and how it specifically benefits the model in this context.

* **Insufficient Discussion of Sparse Adjacency Loss**: While the paper introduces a specially designed sparse adjacency loss, the discussion surrounding its design and impact on model performance is limited. Sparse adjacency loss is an important component, especially given the sparsity of RNA contact maps. Understanding how this loss function influences the model's predictions could provide valuable insights. The authors could expand on why this specific loss was chosen, its advantages over standard loss functions, and its contribution to handling sparse data effectively.

* **Need for Additional Baselines on Synthetic Data**: When comparing results on synthetic data, the paper only uses one baseline method, AlphaFold 3 (AF3), a 3D structure prediction model. In this comparison, the authors first predict the tertiary structure with AF3 and then extract the secondary structure, which may not provide an entirely adequate baseline. Although AF3 is considered state-of-the-art for 3D structure prediction, this does not necessarily imply it is optimal for secondary structure prediction, as the paper only claims AF3’s superiority in 3D tasks. Including additional commonly used secondary structure prediction models as baselines would strengthen the evaluation and provide a more comprehensive comparison.

**Questions:**

I have no question.

---

> ### Author Response · Authors · 2024-11-15
> **Response to Reviewer ULQ7**
>
> Dear Reviewer ULQ7,
>
> Thanks for the valuable feedback. We will address all your concerns in the following:
>
> >**Weaknesses:**
> >**Lack of Explanation for Column and Row Axial Attention:** One limitation of the paper is the insufficient discussion or justification for using column and row axial attention in RNAformer. This attention mechanism is applied in each RNAformer block to process the 2D latent representation of the RNA sequence. In certain models, such as MSA Transformer (https://www.biorxiv.org/content/10.1101/2021.02.12.430858v3), row and column attention is intuitive due to the distinct structural roles of rows and columns in the input data. However, in RNAformer, there is no inherent difference between rows and columns, as the input is a single RNA sequence without a natural row-column structure. Additionally, the output contact map is a symmetric matrix, where rows and columns are effectively equivalent. The authors may want to clarify why this architecture was chosen and how it specifically benefits the model in this context.
>
> On the one hand, we need an architecture that can deal with various lengths and is independent of the input length since we consider RNA sequence lengths from 33 to 200nt. The common usage of CNN-based architectures  (see e.g. [8]) is suboptimal since the receptive field depends on the input size, which is also true for U-Net architectures (see e.g. [1,6]). Only self-attention can deal with arbitrary input lengths while maintaining an input-independent full receptive field.
>
> On the other hand, we focus on the prediction of the adjacency matrix of base pairs because only the adjacency matrix representation can capture both, pseudoknots and multiplets (which appeared in real-world data in over 50% of the RNAs). To do so, we decided to have an architecture with a 2D latent representation (3D tensor with LxLxD shape with Length, and Dimension) that subsequently refines the adjacency matrix in the latent space. This also increases the interpretability of our model since we already model the final output format in the latent space, see Figure 4 and Appendix J.
>
> Since we can’t use fully CNN-based architectures as mentioned before, we have to use attention. A full attention on a 2D latent space would require a 3D attention matrix which is infeasible in terms of memory requirements. Therefore we disentangle the attention in a row- and column-wise attention similar to the pair matrix processing in the Evoformer (AlphFold2).  However, convolutions are well suited to capture local geometries. Therefore, one novel aspect of the RNAformer architecture is that it uses a single 3x3 convolutional layer instead of a feed-forward MLP (e.g. in AlphaFold2) to capture local structures.
>
> As you mentioned the predicted adjacency matrix (contact map) is symmetric. We leverage the symmetric structure by averaging the upper and lower triangular pairings to create a set of the final base pairing predictions. Does this clarify our architecture decision and address your concerns?
>
> >**Insufficient Discussion of Sparse Adjacency Loss:** While the paper introduces a specially designed sparse adjacency loss, the discussion surrounding its design and impact on model performance is limited. Sparse adjacency loss is an important component, especially given the sparsity of RNA contact maps. Understanding how this loss function influences the model's predictions could provide valuable insights. The authors could expand on why this specific loss was chosen, its advantages over standard loss functions, and its contribution to handling sparse data effectively.
>
> The ratio of base pairs to unpaired bases in the adjacency matrix is highly imbalanced. The number of possible base pairs scales linearly with the sequence length but the number of possible pairings in the adjacency matrix quadratically, resulting in a sparse 2D representation. To address this sequence length-dependent imbalance, we introduce the sparse adjacency loss. Given an adjacency matrix, the loss calculation depends on 1) all base pairs and their surrounding regions (all entries in the proximity of 3), since the local structures are very important for the final secondary structure prediction. 2) unpaired bases that are randomly chosen for a fixed ratio of paired/unpaired positions as opposed to a normal loss that would be calculated for every entry in the adjacency matrix. As a result, we ignore more positions in the adjacency matrix with increasing sequence length, addressing the length-dependent imbalance.  An example of how using sparse adjacency loss affects the adjacency matrix is presented in Appendix A.

---

> ### Author Response · Authors · 2024-11-15
>
> >**Need for Additional Baselines on Synthetic Data:** When comparing results on synthetic data, the paper only uses one baseline method, AlphaFold 3 (AF3), a 3D structure prediction model. In this comparison, the authors first predict the tertiary structure with AF3 and then extract the secondary structure, which may not provide an entirely adequate baseline. Although AF3 is considered state-of-the-art for 3D structure prediction, this does not necessarily imply it is optimal for secondary structure prediction, as the paper only claims AF3’s superiority in 3D tasks. Including additional commonly used secondary structure prediction models as baselines would strengthen the evaluation and provide a more comprehensive comparison.
>
> Synthetic data has only been used to show that the RNAformer architecture can effectively learn the biophysical model (i.e. RNAfold) of RNA folding (Section 4.1). Experimental data from PDB, on the other hand, has been used in the experiments with data homology (Section 4.2). The results for fine-tuning on homologous data are presented in Table 2 and on non-homologous data in Table 3. So the comparison to AF3 (Table 2) is performed on known RNA secondary structure data from PDB. The same test data is used in subsequent experiments when compared to the other methods (i.e. we only use different training sets to allow more or less data homologies). Also, while we agree that 3D predictions are different from secondary structure predictions, we still think that the comparison is reasonable since we use the same tool for obtaining secondary structure information from both the ground truth and AF3 predictions (DSSR; commonly used to obtain secondary structure information from 3D RNA data) and because strong 3D structure predictions should also provide good secondary structures.
>
> Since the experiment is performed on potentially homologous data to allow for a fair comparison (because AF3 was trained without any strong homology criteria)it is not further highlighted as a SOTA result in the manuscript.
>
> We hope we have clarified your questions and would be happy to answer any further questions. If we have addressed your concerns, we would be very thankful if you considered raising your score.

---

> ### Comment · Reviewer_ULQ7 · 2024-11-18
>
> Thanks for your reply. The explanation for Sparse Adjacency Loss and Synthetic Data looks good to me. However, I still have doubts about Axial-Attention. My concern is not about why to use a transformer/attention, but rather why Axial-Attention specifically.
>
> If the input were MSA data, then using Axial-Attention makes sense to me because the input is inherently a matrix. However, your model currently takes a single sequence as input, and I believe standard attention should be sufficient to handle this input.
> For example, in a similar task in protein modeling called contact prediction, the goal is to predict which residues in a given sequence have contacts, with the input being a sequence (not MSA) and the output being a symmetric
> 𝐿
> ×
> 𝐿
>  contact map (where
> 𝐿
>  is the sequence length). A classic baseline for this task is TAPE (https://arxiv.org/abs/1906.08230), which uses a standard transformer model and can also predict contact maps.
>
> What advantages does using Axial-Attention have over a standard transformer in this task?

---

> > ### Author Response · Authors · 2024-11-19
> >
> > Thanks for your response! Since we input a sequence and output an adjacency matrix, our model is a function that describes a 1D to 2D mapping. The extension from a 1D to a 2D representation could either happen in the output of the model (such as in TAPE, E2Efold, ProbTransformer), or in the input by creating a 2D latent space based on the input sequence (e.g. by broadcasting) and processing the 1D input into a 2D representation, (e.g. with axial attention, such as in RNAformer or AlphaFold2 Evoformer).
> > We chose the second variant because our final predictive representation is the adjacency matrix of the RNA secondary structure. This is required to allow for the prediction of pseudoknots and multiplets. With this approach, the model can successively refine its latent representation through each layer until finally outputting the structure’s adjacency matrix representation.
> >
> > For clarification, pseudoknots can also be represented as a 1D sequence (e.g., “(..[....)]” ) using an extended vocabulary, but then we need to know the maximal complexity of the nesting beforehand. With a 2D representation, we don’t need to take care of the level of nesting.
> > A multiplet connection on the other hand is a connection of one base with two or more other bases instead of one. This kind of connection can hardly be represented in 1D. Again a 2D adjacency matrix does not have these limitations.
> >
> > As conceptual motivation: For a human, it is hard to represent pseudoknots and multiplets in a 1D representation. Similarly, it might be hard for a model to represent all this information in a 1D latent representation, while a 2D representation gives the model more room to learn a good representation of the RNA secondary structure in each network layer.
> >
> > In preliminary experiments, we tested the broadcasting from 1D to 2D in the input, in the middle of the model (first vanilla transformer, then broadcasting and axial attention layers), and in the model's output (such as TAPE). We found that the sooner we broadcast the better the prediction performance. We also tried to use the “triangular self-attention” from AlphaFold2 Evoformer but didn’t achieve significantly better results compared to standard axial attention.
> >
> > In conclusion, the advantage of axial attention is that we directly model the base-parings in the latent space in a format that can represent pseudoknots and multilets, which will also be used for the final prediction output of the base pair adjacency matrix and is interpretable in the intermediate layers.
> >
> > We hope this addresses your question regarding the usage of axial attention. If it does, we kindly ask you to consider reflecting this in your scoring, potentially supporting the paper's acceptance.
> > If you have any further questions or need additional clarification, we would be more than happy to provide it. Thank you for your thoughtful review!

---

> ### Author Response · Authors · 2024-11-25
> **Updated manuscript and title**
>
> Dear Reviewer ULQ7,
>
> We hope this message finds you well. Did you have a chance to read our responses?
> In the meantime, we changed the title to “A simple yet effective model for homology-aware RNA secondary structure prediction” to reduce the focus on axial-attention and emphasize the contribution of the paper. We further updated our manuscript according to the feedback from the reviews. We added a paragraph on the motivation for the usage of the axial-attention. We kindly ask if you could reconsider your score or let us know if further clarifications are needed.
>
> Thank you and kind regards.

---

> ### Comment · Reviewer_ULQ7 · 2024-11-26
>
> The authors' response has effectively addressed my concerns. I will keep my score, as my initial scoring was based on the assumption that the authors would provide a satisfactory explanation for my questions, which they have now confirmed.

---

### Official Review · Reviewer_qpbb · 2024-11-10

**Soundness:** 2
**Presentation:** 3
**Contribution:** 2
**Rating:** 3
**Confidence:** 3

**Summary:**

The paper introduces RNAformer, a deep learning model designed for predicting RNA secondary structures from a single sequence without additional requirements like multiple sequence alignments or thermodynamic parameters. The model uses an axial-attention mechanism that is independent of input size, making it suitable for RNA sequences of varying lengths. RNAformer is trained on a homology-aware data pipeline that addresses sequence and structure similarities to ensure a clean split between training and test data.

**Strengths:**

RNAformer is a single deep learning model that relies solely on RNA sequence input and does not require additional information like multiple sequence alignments (MSAs), embeddings, or ensemble techniques. Also, the Axial-Attention makes the model to be efficient to the length of RNA sequence.

**Weaknesses:**

The novelty is limited, where the key technique Axial-Attention has already been proposed in 2019. The authors do not appear to have designed the algorithm specifically for RNA data. Rigorous data splitting is too common sense to be a significant contribution to be published at an AI conference. Compared to baselines, it seems to be have weak constraints on RNA secondary structure, such as ES and TP post processing.

**Questions:**

Q1: How can your algorithm ensure the RNA secondary structure constraints, such as symmetry, no sharp loop within three bases, and so on. Refer to [1].

Q2: Could you compare your algorithm to recent studies[1,2,3]?

Q3: Regarding the Axial-Attention, what you have done to make it spcified to your problem? What is the difference when applying it to image data and RNA data?

Q4: What is the distribution of sequence length? What are the average and maximum lengths?

[1] Tan, Cheng, et al. "Deciphering RNA Secondary Structure Prediction: A Probabilistic K-Rook Matching Perspective." Forty-first International Conference on Machine Learning.

[2] Chen, Xinshi, et al. "RNA secondary structure prediction by learning unrolled algorithms." arXiv preprint arXiv:2002.05810 (2020).

[3] Jung, Andrew J., et al. "RTfold: RNA secondary structure prediction using deep learning with domain inductive bias." The 2022 ICML Workshop on Computational Biology. Baltimore, Maryland, USA. 2022.

---

> ### Author Response · Authors · 2024-11-15
> **Response to Reviewer qpbb**
>
> Dear Reviewer qpbb,
>
> Thanks for the valuable feedback. We would like to mention that our model achieved SOTA performance on the PDB data, the only data collection derived from experimentally (in-vitro) validated RNA 3D structures. This is not part of your summary or listed in the strengths, but we find this a crucial aspect of our work, making the RNAformer a practical tool for biologists. We will address all your concerns and provide answers to questions in the following:
>
> >Weaknesses:
> >The novelty is limited, where the key technique Axial-Attention has already been proposed in 2019. The authors do not appear to have designed the algorithm specifically for RNA data.
>
> The novelty of our work is twofold. First, we prove that a simple axial-attention-based model is capable of achieving SOTA performance on RNA secondary structure prediction. We show that there is no need for MSA, ensembles of architectures, additional features or constraints, or sophisticated pre- and post-processing.
>
> Furthermore, we disagree with the reviewer that the RNAformer architecture is not specifically designed for RNA data. On the one hand, we need an architecture that can deal with various lengths and is independent of the input length since we consider RNA sequence lengths from 33 to 200nt. The common usage of CNN-based architectures  (see e.g. [8]) are suboptimal since the receptive field depends on the input size, which is also true for U-Net architectures (see e.g. [1,6]). Only self-attention can deal with arbitrary input lengths while maintaining an input-independent full receptive field.
>
> On the other hand, we focus on the prediction of the adjacency matrix of base pairs because only the adjacency matrix representation can capture both, pseudoknots and multiplets (which appeared in real-world data in over 50% of the RNAs). To do so, we decided to have an architecture with a 2D latent representation (3D tensor with LxLxD shape with Length, and Dimension) that subsequently refines the adjacency matrix in the latent space. This also increases the interpretability of our model since we already model the final output format in the latent space, see Figure 4 and Appendix J.
>
> Since we can’t use fully CNN-based architectures as mentioned before, we have to use attention. A full attention on a 2D latent space would require a 3D attention matrix which is infeasible in terms of memory requirements. Therefore we disentangle the attention in a row- and column-wise attention similar to the pair matrix processing in the Evoformer (AlphFold2).  However, convolutions are well suited to capture local geometries. Therefore, one novel aspect of the RNAformer architecture is that it uses a single 3x3 convolutional layer instead of a feed-forward MLP (e.g. in AlphaFold2) to capture local structures. Another problem-specific novelty is the Sparse Adjacency Loss which we describe in Section 2.2 and is crucial for the SOTA performance.
>
> Secondly, we present a novel data processing pipeline that ensures no data homologies between training and test sets. Various biologists [9,10,11] have raised concerns that deep learning-based models do not consider this (e.g. using bpRNA or ArchiveII data to evaluate). We address this criticism and show that our model can achieve SOTA performance while being homology-aware.
>
> An alternative title for our paper could be “A simple yet efficient deep-learning model for RNA secondary structure prediction with full homology awareness”. Does this clarify our architecture design decision and address your concerns?

---

> ### Author Response · Authors · 2024-11-15
>
> >Rigorous data splitting is too common sense to be a significant contribution to be published at an AI conference.
>
> We think it’s exactly the opposite. It is very important to raise awareness of the data homology problem in RNA at AI conferences. Recently multiple groups with biological backgrounds (see [9,10,11]) named caveats about the usage of deep learning in RNA structure biology. The main concern is insufficiently curated test data.
>
> In particular, the usage of test data sets which are not based on experimentally validated RNA structures and homology unaware splits between train and test data cause skepticism and harm the reliability and practical relevance of the developed models.
>
> For instance, training and evaluating on the RNAStralign and ArchiveII datasets as provided by Chen et al. 2020 [2] was shown to lead to overfitting and bad generalization capabilities (see [13]), and the bpRNA based TR0 and TS0 split as provided by Singh et al., 2019 [5], are known for homologies (see [9]). Nevertheless, we recently observed many approaches published at the latest AI conferences and top-tier journals that do not consider a rigorous data split, such as:
> - RFold (ICML 2024) [1]
> - E2Efold (ICLR 2020) [2]
> - RTfold (WCB@ICML2022) [3]
> - Probabilistic Transformer (NeurIPS 2022) [4]
> - SPOT-RNA (Nature Communications) [5]
> - UFold (NAR) [6]
> - RNA-FM (arXiv) [7]
> - (AlphaFold3 (Nature 2024); 3D prediction, yes, but still)
>
>
> We provide the first evaluations of a deep learning method for RNA secondary structure prediction on a non-homologous data split for experimentally validated structures.
>
> This sets a new standard for RNA data processing which allows deep learning practitioners to thoroughly evaluate their methods. We consider this a major contribution to the community and think that this is exactly the next step needed to bring deep-learning-based RNA structure prediction into production.
>
> >Compared to baselines, it seems to be have weak constraints on RNA secondary structure, such as ES and TP post processing.
>
> The reviewer is right that we don’t have constraints on the RNA secondary structure but achieve state-of-the-art performance without the need for sophisticated pre- or post-processing, nor require constraint optimization techniques. In contrast, we learn the secondary structure features directly from the data, which allows us to predict all possible base pairs which is in stark contrast to the work described in [1].
>
> We hope that this answer addresses the reviewers' concerns. However, we are not entirely sure what is meant by ES and TP and would appreciate an explanation if the response is inadequate.
>
> >**Questions:**
>
> >Q1: How can your algorithm ensure the RNA secondary structure constraints, such as symmetry, no sharp loop within three bases, and so on. Refer to [1].
>
> With the RNAformer, we seek to learn RNA secondary structure features in a purely data-driven manner without the need for heuristics and restrictions on the generated structures. For instance, one of the explicit constraints provided in [1] is the restriction to canonical base pairs only. However, the prediction of non-canonical base pairs is one of the major challenges in RNA structure prediction and the lack of such interactions is thus a clear limitation of the algorithm described in [1]. Furthermore, we show with our experiments for learning a biophysical model (Section 4.1) that this constraint can be learned directly from the data as also stated in line 355 of our initial submission:
>
> “In accordance with the underlying data generated with RNAfold, which only contains canonical base pairs, the RNAformer does not predict any non-canonical base pairs across three independent runs.”
>
> Similarly, the constraints formulated in [1] result in structure predictions without  base multiplets, an important pattern e.g. for the formation of G-quadruplex structures and highly abundant in our PDB-derived test data.
>
> Generally, we do not observe systematic errors (like e.g. lack of symmetry or any other recurring errors) for the predictions of RNAformer throughout our experiments. Further, symmetry constraints as well as a constraint on loop sizes could easily be implemented on top of RNAformer in a small post-processing pipeline in case one would like to build a product. Does this answer your question regarding constraints?

---

> ### Author Response · Authors · 2024-11-15
>
> >Q3: Regarding the Axial-Attention, what you have done to make it spcified to your problem? What is the difference when applying it to image data and RNA data?
>
> We outline the motivation for using the axial-attention above. Another aspect is that there are only 3400 high-quality training samples for RNA secondary structures and a design decision to adapt our architecture to this data could easily lead to an overfitting on the architecture level. Therefore, we tried to keep the architecture as lean as possible. However, we tried to use the “triangular multiplicative update” similar to AlphaFold but didn’t achieve significantly better results.
>
> Another problem-specific novelty is the Sparse Adjacency Loss. We use a masking technique to address the heavy and sequence-length-dependent imbalance between base pairs to no base pairs in the adjacency matrix. The number of possible base pairs scales linearly with the sequence length but the number of possible pairings in the adjacency matrix quadratically, resulting in a sparse 2D representation. Therefore, this loss is crucial to the training success of the RNAformer.
>
> Image data would require a different embedding, a specific head and most vision transformers do not use axial attention due to the high memory consumption but consider an image as a sequence of image patches.
>
> >Q4: What is the distribution of sequence length? What are the average and maximum lengths?
>
> We provide a detailed table with all the length values in Appendix B. For the data derived from 3D RNAs from PDB, the maximum sequence length is 200 nt, with an average length of 57.9nt. Generally, high-quality RNA data is rare, covering roughly 1% of all residues of all known structure data of RNA, DNA, and protein. While there exist longer RNAs in PDB, these belong to a very limited set of RNA families (mainly ribosomal RNAs) which are highly homologous, very long (>2000 nt), and therefore not suitable to improve the prediction of short and mid-length RNA structures.
>
>
> We hope we have clarified your questions and would be happy to answer any further questions. If we have addressed your concerns, we would be very thankful if you considered raising your score. (We have seen jumps from 3 to 8 in the past ;-))
>
> ___
> [1] Tan, Cheng, et al. "Deciphering RNA Secondary Structure Prediction: A Probabilistic K-Rook Matching Perspective." Forty-first International Conference on Machine Learning.
>
> [2] Chen, Xinshi, et al. "RNA secondary structure prediction by learning unrolled algorithms." arXiv preprint arXiv:2002.05810 (2020).
>
> [3] Jung, Andrew J., et al. "RTfold: RNA secondary structure prediction using deep learning with domain inductive bias." The 2022 ICML Workshop on Computational Biology. Baltimore, Maryland, USA. 2022.
>
> [4] Franke, Jörg, Frederic Runge, and Frank Hutter. "Probabilistic transformer: Modelling ambiguities and distributions for RNA folding and molecule design." Advances in Neural Information Processing Systems 35 (2022): 26856-26873.
>
> [5] Singh, Jaswinder, et al. "RNA secondary structure prediction using an ensemble of two-dimensional deep neural networks and transfer learning." Nature communications 10.1 (2019): 5407.
>
> [6] Fu, Laiyi, et al. "UFold: fast and accurate RNA secondary structure prediction with deep learning." Nucleic acids research 50.3 (2022): e14-e14.
>
> [7] Chen, Jiayang, et al. "Interpretable RNA foundation model from unannotated data for highly accurate RNA structure and function predictions." arXiv preprint arXiv:2204.00300 (2022).
>
> [8] Saman Booy, M., Ilin, A., & Orponen, P. (2022). RNA secondary structure prediction with convolutional neural networks. BMC bioinformatics, 23(1), 58.
>
> [9] Flamm, Christoph, et al. "Caveats to deep learning approaches to RNA secondary structure prediction." Frontiers in Bioinformatics 2 (2022): 835422.
>
> [10] Szikszai, Marcell, et al. "Deep learning models for RNA secondary structure prediction (probably) do not generalize across families." Bioinformatics 38.16 (2022): 3892-3899.
>
> [11] Qiu, Xiangyun. "Sequence similarity governs generalizability of de novo deep learning models for RNA secondary structure prediction." PLOS Computational Biology 19.4 (2023): e1011047.
>
> [12] Rivas, Elena, Raymond Lang, and Sean R. Eddy. "A range of complex probabilistic models for RNA secondary structure prediction that includes the nearest-neighbor model and more." RNA 18.2 (2012): 193-212.
>
> [13] Sato, Kengo, Manato Akiyama, and Yasubumi Sakakibara. "RNA secondary structure prediction using deep learning with thermodynamic integration." Nature communications 12.1 (2021): 941.

---

> ### Author Response · Authors · 2024-11-20
>
> Dear Reviewer qpbb,
>
> We thank you once again for your thoughtful feedback. We have done our best to address all your concerns in the rebuttal and would be happy to clarify further if needed. We kindly ask you to reconsider your score in light of our responses or engage with us if further clarifications are required.
>
> Best regards.

---

> ### Author Response · Authors · 2024-11-25
> **Updated manuscript and title**
>
> Dear Reviewer qpbb,
>
> We hope this message finds you well. Did you have a chance to read our responses?
> In the meantime, we changed the title to “A simple yet effective model for homology-aware RNA secondary structure prediction” to reduce the focus on axial-attention and emphasize the contribution of the paper. We further updated our manuscript according to the feedback from the reviews. Here we mainly clarified the contribution of our work and the motivation for the usage of the axial attention.
> We kindly ask if you could reconsider your score or let us know if further clarifications are needed.
>
> Thank you and kind regards.

---

> ### Author Response · Authors · 2024-11-29
> **Gentle follow-up on rebuttal response**
>
> Dear Reviewer qpbb,
>
> I hope this finds you well. I noticed that our rebuttal response from 15th November hasn't received any feedback yet. Given the approaching decision deadline, we wanted to check if you had a chance to review our clarifications and if you have any additional questions we can address. If you have no further concerns, we kindly ask to reconsider your score.
>
> Best regards, The Authors

---

### Author Response · Authors · 2024-12-04
**References for final general response**

[1] Multiple sequence alignment-based RNA language model and its application to structural inference. Nucleic Acids Research, 2024


[2] Busaranuvong, Palawat, et al. "Graph Convolutional Network for predicting secondary structure of RNA." Research Square (2024)


[3] Chen, Chun-Chi, and Yi-Ming Chan. "REDfold: accurate RNA secondary structure prediction using residual encoder-decoder network." BMC bioinformatics 24.1 (2023)


[4] Wei, Sen, and Shengjin Wang. "Structural stability-aware deep learning: advancing RNA secondary structure prediction." Fourth International Conference on Biomedicine and Bioinformatics Engineering (ICBBE 2024). Vol. 13252. SPIE, 2024.


[5] Yang, Enbin, et al. "GCNfold: A novel lightweight model with valid extractors for RNA secondary structure prediction." Computers in Biology and Medicine 164 (2023)


[6] Zhao, Qi, et al. "RNA independent fragment partition method based on deep learning for RNA secondary structure prediction." Scientific Reports 13.1 (2023)


[7] Khrisna, et. al. "The Use of Convolutional Neural Networks for RNA Protein Prediction." 2023 3rd International Conference on Intelligent Cybernetics Technology & Applications (ICICyTA). IEEE, 2023.

[8] Flamm, Christoph, et al. "Caveats to deep learning approaches to RNA secondary structure prediction." Frontiers in Bioinformatics 2 (2022): 835422.

[9] Szikszai, Marcell, et al. "Deep learning models for RNA secondary structure prediction (probably) do not generalize across families." Bioinformatics 38.16 (2022): 3892-3899.

[10] Qiu, Xiangyun. "Sequence similarity governs generalizability of de novo deep learning models for RNA secondary structure prediction." PLOS Computational Biology 19.4 (2023): e1011047.

[11] Rivas, Elena, Raymond Lang, and Sean R. Eddy. "A range of complex probabilistic models for RNA secondary structure prediction that includes the nearest-neighbor model and more." RNA 18.2 (2012): 193-212.

[12] Tan, C., Gao, Z., & Li, S. Z. (2022). RFold: RNA Secondary Structure Prediction with Decoupled Optimization. arXiv preprint arXiv:2212.14041.


[13] Chen, X., Li, Y., Umarov, R., Gao, X., & Song, L. RNA Secondary Structure Prediction By Learning Unrolled Algorithms. In International Conference on Learning Representations.


[14] Jung, A. J., Lee, L. J., Gao, A. J., & Frey, B. J. (2022). RTfold: RNA secondary structure prediction using deep learning with domain inductive bias. In The 2022 ICML Workshop on Computational Biology. Baltimore, Maryland, USA.


[15] Franke, J., Runge, F., & Hutter, F. (2022). Probabilistic transformer: Modelling ambiguities and distributions for rna folding and molecule design. Advances in Neural Information Processing Systems, 35, 26856-26873.


[16] Singh, J., Hanson, J., Paliwal, K., & Zhou, Y. (2019). RNA secondary structure prediction using an ensemble of two-dimensional deep neural networks and transfer learning. Nature communications, 10(1), 5407.


[17] Fu, L., Cao, Y., Wu, J., Peng, Q., Nie, Q., & Xie, X. (2022). UFold: fast and accurate RNA secondary structure prediction with deep learning. Nucleic acids research, 50(3), e14-e14.


[18] Chen, J., Hu, Z., Sun, S., Tan, Q., Wang, Y., Yu, Q., ... & Li, Y. (2022). Interpretable RNA foundation model from unannotated data for highly accurate RNA structure and function predictions. arXiv preprint arXiv:2204.00300.


[19] Abramson, J., Adler, J., Dunger, J., Evans, R., Green, T., Pritzel, A., ... & Jumper, J. M. (2024). Accurate structure prediction of biomolecular interactions with AlphaFold 3. Nature, 1-3.

---

### Author Response · Authors · 2024-12-04
**Our valuable contribution and good reasons to accept**

Dear Reviewers, Area Chairs, Senior Area Chairs, and Program Committee,

We respectfully disagree with some aspects of the current rating and would like to highlight key points why **we see our work as a valuable contribution and good reasons to accept** our RNA secondary structure prediction approach:

- We introduce the RNAformer based on axial-attention and argue that **attention is better suited than the commonly used CNNs [1-7] for processing sequences with varying length** such as RNA and show that such a **lean architecture built from existing components achieves SOTA results** without the need for ensamples, pre/post processing or additional MSA.
- We **address the caveats raised by the RNA community** [8-11] by introducing a novel data pre-processing pipeline that allows us to split training and test data while accounting for homologies. This homology-aware data split and evaluation is not common and many approaches still evaluate on homologous data [3, 5, 6, 12-19].
- We show the importance of removing homologies between training and test data in an experiment where we **outperform AlphaFold 3 in terms of secondary structure accuracy** (more precisely, we show that AlphaFold 3 performs worse on samples where there is no MSA available while RNAformer’s performance is constant, consistently outperforming AlphaFold 3).
- Even so, we consider homologies, RNAformer achieves SOTA results and **outperforms models that use homologous data or additional information such as MSA on experimentally derived PDB** data.

We thank the reviewers for their feedback and discussion. We responded to all reviews, answered all questions, and updated the manuscript according to the feedback from the reviewers. We summarize the reviewer interaction as follows:

- Reviewer qpbb (rating 3, confidence 3) didn’t acknowledge the SOTA results, emphasized the **lean, efficient, and single sequence-based design** but criticized the lack of a novel architecture specific to RNA secondary structure, and claimed the RNA data split was common sense. We explained that our point is that current deep learning architectures are already sufficiently well suited to achieve SOTA results and explained the homology issues in recent related works. We answered all questions but we got **zero responses**.
- Reviewer ULQ7 (rating 6, confidence 4) emphasizes the **“model simplicity”, “well-designed experiments with superior performance”, and the “rigorous dataset construction”**. The reviewer named weaknesses that we addressed satisfactorily according to the reviewer's response.
- Reviewer BzZh (rating 5, confidence 4) didn’t write a summary but **named as the strength that the data pipeline addresses critical aspects of the research community, the biophysical experiment and that the “experimental results are useful”**. The reviewer criticized the lack of a novel architecture specific to RNA secondary structure, and the similarity in the synthetic dataset. Again, we explained that our point is that current deep learning architectures are already good enough to achieve SOTA results, addressed the data concern, and answered the questions but we got **zero responses**.
- Reviewer aKqc  (rating 3→6, confidence 3) was most active in the discussion and gave good feedback, thanks again. The reviewer emphasizes **“model’s performance and scalability”** and named as weakness the motivation for the axial attention, lack of specific architecture for RNA secondary structure, and that the methodology of the datasets needs clarification. We addressed all these points and the reviewer increased the rating from recject to minor accept.
- Reviewer UEd4  (rating 5, confidence 3) emphasized the **scalability, efficiency, and homology-aware approach and that our model “outperforming other models in accuracy and generalization”**. The reviewer criticized mainly the manuscript by asking for better explanation and naming issues. The reviewer sees a lack of novelty on the algorithmic level, but the reviewer also admits that they are “not particularly knowledgeable about the application domain” and that they will follow other reviewers’ decisions.

In total, the reviewers highlighted RNAformer's scalability, efficient design, the homology-aware data processing, and ability to achieve strong performance using sequence data alone as its major strengths. **We addressed all concerns raised by the reviewers**, answered all questions, and updated the manuscript and title according to the feedback from the reviewers.

@AC: Due to the lack of the two remaining critical reviewers' responses, we respectfully ask you to **make your own assessment** of our work for the meta-review, **rather than relying on the current scores**. Again, we believe **our work makes a valuable contribution** to the field.

Thank you for your consideration and your service to ICLR.

Best regards, The Authors

---

### Meta-Review · Area_Chair_wmxv · 2024-12-17

**Metareview:**

This paper proposes RNAformer, a pretty lean approach for RNA secondary structure prediction, relying primarily on a single sequence and axial-attention. The authors claim that using a homology-aware data pipeline and avoiding MSAs or complex ensembles, their model still achieves what they call SOTA performance on experimentally validated PDB structures.

The authors were active in responding to the reviews, going through all comments in detail. I personally highly appreciate this!
They clarified the reasoning behind their data splits, pointed out that previous methods often evaluated on homologous datasets, and explained how axial-attention can handle sequence lengths and capture pairwise structure information.

After the rebuttal phase, not all reviewers changed their view. One reviewer (aKqc) initially skeptical about the motivation for axial-attention, ended up slightly more positive after the authors provided extra experiments and data. Another reviewer (ULQ7) remained supportive, acknowledging the model’s simplicity and the careful homology-aware evaluation. On the other hand, a couple of reviewers (qpbb and UEd4) felt that the algorithmic novelty was too limited, even after multiple attempts by authors to highlight the training data pipeline and sparse adjacency loss. They never re-engaged positively. Personally I am also on the side of thinking the novelty can be better justified or improved.

I think the paper’s main contributions—showing strong results without MSAs and carefully ensuring no test-train homology—is relevant and addresses key criticisms in RNA structure prediction research.

**Additional Comments On Reviewer Discussion:**

As discussed above directly.

---

### Decision · Program_Chairs · 2025-01-22

Reject